# Structure and elevator mechanism of the mammalian sodium/proton exchanger NHE9

Iven Winkelmann[1,†], Rei Matsuoka[1,†], Pascal F Meier[1,†], Denis Shutin[2], Chenou Zhang[3], Laura Orellana[1], Ricky Sexton[3], Michael Landreh[4] [ID], Carol V Robinson[2], Oliver Beckstein[3] & David Drew[1,*] [ID]

## Abstract

Na$^+$/H$^+$ exchangers (NHEs) are ancient membrane-bound nanomachines that work to regulate intracellular pH, sodium levels and cell volume. NHE activities contribute to the control of the cell cycle, cell proliferation, cell migration and vesicle trafficking. NHE dysfunction has been linked to many diseases, and they are targets of pharmaceutical drugs. Despite their fundamental importance to cell homeostasis and human physiology, structural information for the mammalian NHEs was lacking. Here, we report the cryogenic electron microscopy structure of NHE isoform 9 (SLC9A9) from *Equus caballus* at 3.2 Å resolution, an endosomal isoform highly expressed in the brain and associated with autism spectrum (ASD) and attention deficit hyperactivity (ADHD) disorders. Despite low sequence identity, the NHE9 architecture and ion-binding site are remarkably most similar to distantly related bacterial Na$^+$/H$^+$ antiporters with 13 transmembrane segments. Collectively, we reveal the conserved architecture of the NHE ion-binding site, their elevator-like structural transitions, the functional implications of autism disease mutations and the role of phosphoinositide lipids to promote homodimerization that, together, have important physiological ramifications.

**Keywords** membrane protein; SLCA9; pH regulation; sodium/proton exchanger; structure

**Subject Categories** Membrane & Trafficking; Structural Biology

The EMBO Journal (2020) 39: e105908

## Introduction

Na$^+$/H$^+$ exchangers (NHEs) are ion transporters found in all kingdoms of life (Orlowski & Grinstein, 2004; Donowitz *et al*, 2013; Fuster & Alexander, 2014; Pedersen & Counillon, 2019). They directly couple the transfer of protons across biological membranes to the counter-transport of Na$^+$/Li$^+$/(K$^+$), a mechanism first proposed in bacteria (West & Mitchell, 1974) and later observed in isolated, rat kidney vesicles (Murer *et al*, 1976). In mammals, there are 13 distinct NHE orthologues that are thought to perform electroneutral (1:1) ion-exchange: NHE1-9 also known as SLC9A1-9, NHA1-2 also known as SLC9B1-2 and sperm-specific NHE also known as SLC9C (Brett *et al*, 2005; Fuster & Alexander, 2014; Pedersen & Counillon, 2019). NHEs differ in substrate preferences, kinetics and tissue localizations (Pedersen & Counillon, 2019). NHE1, for example, is ubiquitously expressed in the plasma membrane of most tissues, and its major physiological role is the regulation of intracellular pH and cell volume (Slepkov *et al*, 2007; Pedersen & Counillon, 2019). NHE3, on the other hand, is highly expressed in the intestine and kidneys and is important for Na$^+$ reabsorption and acid–base homeostasis (Zachos *et al*, 2005; Donowitz *et al*, 2009). Other NHE isoforms, such as NHE6, NHE7, NHE8 and NHE9, are critical for the maintenance and regulation of organellar and endosomal pH, which in turn are linked to a multitude of physiological functions (Brett *et al*, 2002; Orlowski & Grinstein, 2007; Pedersen & Counillon, 2019).

Dysfunction of NHEs has been linked to many diseases such as cancer, hypertension, heart failure, diabetes and epilepsy (Fuster & Alexander, 2014; Ueda *et al*, 2017). In particular, NHEs are prime drug targets for cancer therapies (Stock & Pedersen, 2017; Pedersen & Counillon, 2019), since tumour cells typically upregulate NHE expression to re-alkalinize intracellular pH in response to the "Warburg effect" (Cardone *et al*, 2005; Parks *et al*, 2013), i.e. as metabolic preference for oxidative glycolysis leads to intracellular acidification. Consequently, many cancer cells are highly dependent on NHE activity, and their inhibition or knockdown interferes with cancer development (Cardone *et al*, 2005; Stock & Pedersen, 2017; White *et al*, 2017). NHE1 has further been targeted in heart disease (Odunewu-Aderibigbe & Fliegel, 2014), since NHE1 inhibition protects the myocardium against ischaemic, reperfusion injury and heart failure. Although clinical trials of an NHE1 inhibitor were discontinued, animal models hold promise that NHE1 inhibition could result in an effective therapeutic (Karmazyn, 2013). Since NHE3 has been linked directly to blood volume and pressure (Alexander & Grinstein, 2006), NHE3 is targeted as an avenue to treat hypertension (Linz *et al*, 2016); so far, this is yet to bear fruit,

1   Department of Biochemistry and Biophysics, Stockholm University, Stockholm, Sweden
2   Department of Chemistry, University of Oxford, Oxford, UK
3   Department of Physics, Center for Biological Physics, Arizona State University, Tempe, AZ, USA
4   Department of Microbiology, Tumor and Cell Biology, Karolinska Institute, Stockholm, Sweden
    *Corresponding author. Tel: +46 816 2295; E-mail: d.drew@dbb.su.se
    †These authors contributed equally to this work

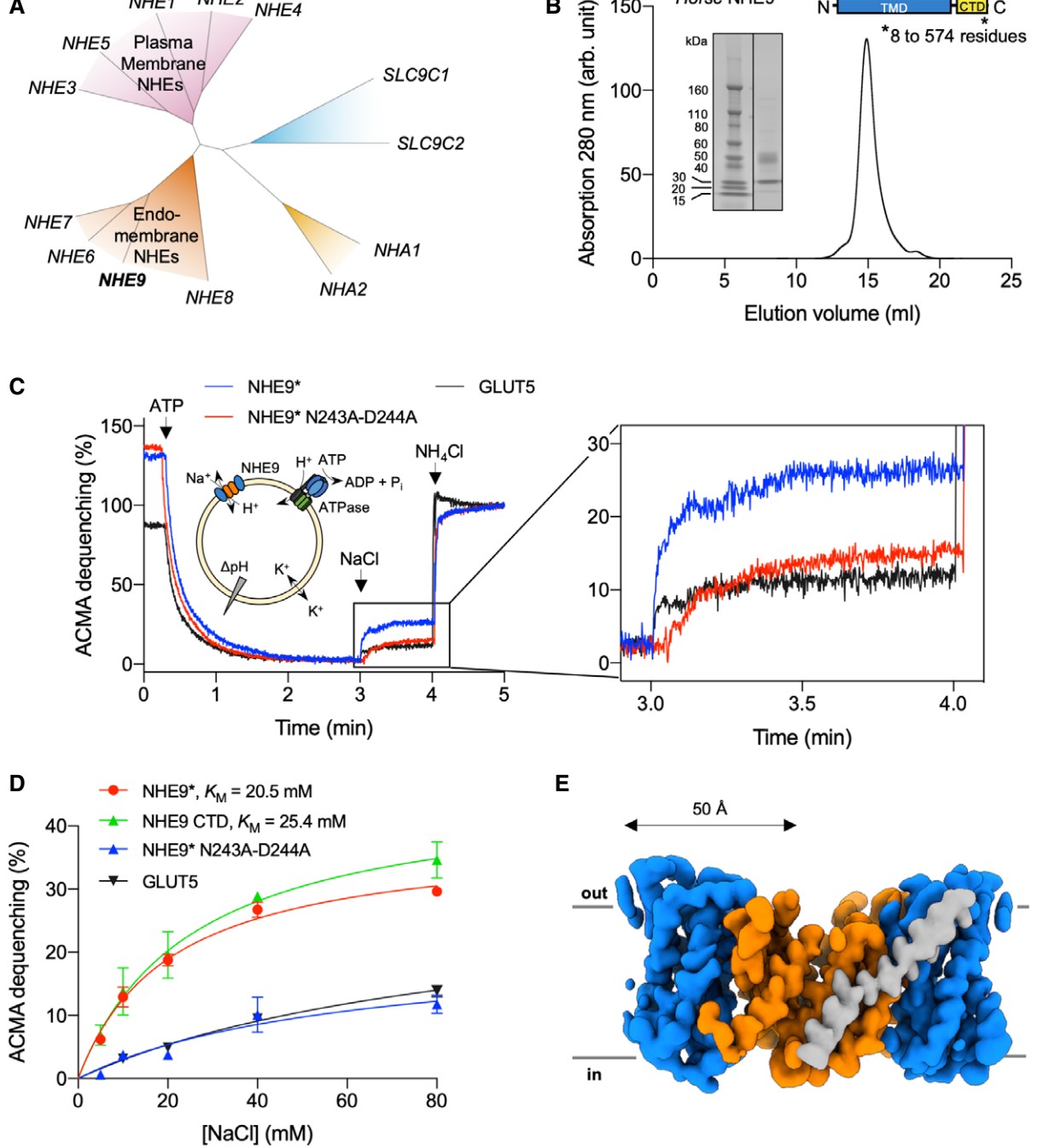

**Figure 1.  *Horse* NHE9 functional characterization and cryo-EM structure[‡].**

A   Phylogenetic tree of canonical *human* NHE1-9 (SLC9A1-9) cluster into those that localize predominantly to either the plasma membrane (pink) or endomembrane (orange); for completeness, more distantly related non-canonical *human* NHE members SLC9B1-2 (NHA1-2) (yellow) and SLC9C (blue) are also shown as labelled.

B   SEC trace and SDS–PAGE of purified NHE9* (residues 8 and 574 out of 644) as depicted by schematic.

C   Activity of NHE9* co-reconstituted with ATPase into liposomes to mimic the *in vivo* situation (schematic). Representative ACMA fluorescence traces of liposome reconstituted NHE9* (blue), NHE9* double-mutant N243A-D244A (red) and rat fructose transporter GLUT5 (black). ATP-driven H[+] pumping establishes a ΔpH (0–3 min). H[+] efflux is initiated by the addition of 40 mM NaCl, and subsequent addition of NH₄Cl (4 min) collapses the proton gradient.

D   Michaelis–Menten kinetics for NHE9* (red), NHE9 ΔCTD (green), N243A-D244A (blue) and rat GLUT5 (black) as detected by ACMA dequenching following substrate addition. In all experiments, error bars, s.e.m.; $n$ = 3 technical repeats. The apparent $K_M$ values are an average from $n$ = 3 separate protein reconstitutions.

E   Cryo-EM density map of the NHE9 ΔCTD homodimer with the 6-TM core transport domains (coloured in blue), the dimer domain (coloured in orange) and the linker helix (coloured in grey).

[‡]Correction added on 15 December 2020, after first online publication: In the previous version of Figure 1C, the colour scheme in the magnification had been inverted between NHE9 WT and NHE9 N243A-D244A. Additionally, the label in Figure 1C for NHE9 WT was updated to NHE9*.

although an NHE3 inhibitor was awarded FDA-approval for irritable bowel syndrome (Siddiqui & Cash, 2020). The endosomal NHE6 and NHE9 are the only isoforms known, to date, with disease-associated mutations (Fuster & Alexander, 2014). More specifically, human disease mutations of NHE6 are associated with an Angelman syndrome-like disorder (Gilfillan *et al*, 2008) and NHE9 to neurological disorders such as familial autism, ADHD and epilepsy (Kondapalli *et al*, 2013, 2014; Ullman *et al*, 2018).

Despite the clear importance of NHE function to human physiology and drug development, their structure and the molecular details of their ion-exchange mechanism have been lacking. Based on substantial biochemical data, all NHEs are thought to form physiological active homodimers (Brett *et al*, 2005; Fuster & Alexander, 2014; Pedersen & Counillon, 2019), with the respective monomers consisting of a transporter module that performs ion-exchange, and a C-terminal non-membranous cytosolic domain of varying length ~ 125–440 residues, which regulates ion-exchange activity (Fuster & Alexander, 2014; Pedersen & Counillon, 2019). The transport module shares ~18–25% sequence homology to bacterial $Na^+/H^+$ antiporters harbouring the "NhaA-fold", so-named after the first crystal structure obtained from *Escherichia coli* (Brett *et al*, 2005; Padan, 2008; Fliegel, 2019). The NhaA crystal structure and more recent structures from other bacterial homologues (Hunte *et al*, 2005; Lee *et al*, 2013; Paulino *et al*, 2014; Wöhlert *et al*, 2014) have shown that the transporter module consists of two distinct domains, a dimerization domain and an ion-transporting (core) domain, made up of six transmembrane (TM) segments. The 6-TM core domain is thought to undergo global, elevator-like structural transitions to translocate ions across the membrane against the anchored dimerization domain (Lee *et al*, 2013; Coincon *et al*, 2016; Drew & Boudker, 2016; Okazaki *et al*, 2019).

Due to their highly dynamic nature, the poor stability of detergent-solubilized NHEs has been a bottleneck for structural studies. Here, we focused our structural efforts on NHE9 since this isoform possesses one of the shortest C-terminal regulatory domains, simplifying mechanistic understanding, and an atomic model would allow us to interpret human disease mutations. Whilst the full physiological role of NHE9 still remains uncertain, NHE9 activity is important for regulating vesicular trafficking and turnover of the synaptic membranes, by the fine-tuning of endosomal pH (Kondapalli *et al*, 2015). In mouse hippocampal neurons, the absence of NHE9 causes impaired synaptic vesicle exocytosis by reducing presynaptic $Ca^{2+}$ entry as a consequence of altered luminal pH (Ullman *et al*, 2018). NHE9 is also highly expressed in glioblastoma multiforme (GBM), the most common brain tumour, as endolysosomal pH is critical for epidermal growth factor (EGFR) sorting and turnover (Kondapalli *et al*, 2015).

## Results

Out of 13 candidates, *horse* NHE9, which shares 95% sequence identity with *human* NHE9, was identified to be the most detergent stable using fluorescence-based screening methods in *Saccharomyces cerevisiae* (Materials and Methods, Figs 1A and B and EV1, and Appendix Fig S1). Purified *horse* NHE9* (residues 8–574) was

reconstituted into liposomes together with $F_0F_1$-ATP synthase to mimic the *in vivo* co-localization of NHE9 with the endosomal V-ATPase (Kondapalli *et al*, 2015). Proton efflux was monitored in response to the addition of $Na^+$, performed in the presence of valinomycin, which was included to eliminate efflux against a membrane

**Table 1. Data collection, processing and refinement statistics of NHE9 structures[§].**

| | NHE9* (EMDB-11066) (PDB 6Z3Y) | NHE9 ΔCTD (EMDB-11067) (PDB 6Z3Z) |
|---|---|---|
| Data collection and processing statistics | | |
| Magnification | 165,000 | 165,000 |
| Voltage (kV) | 300 | 300 |
| Electron exposure (e⁻/Å²) | 80 | 80 |
| Defocus range (μm) | 0.7–2.5 | 0.7–2.5 |
| Pixel size (Å) | 0.83 | 0.83 |
| Symmetry imposed | C1 | C1 |
| Initial particle images (no.) | 2,781,170 | 1,629,483 |
| Final particle images (no.) | 139,511 | 595,024 |
| Map resolution (Å) | 3.51 | 3.19 |
| FSC threshold | 0.143 | 0.143 |
| Map resolution range (Å) | 3.5–4.9 | 3.1–4.1 |
| Refinement | | |
| Initial model used (PDB code) | 4cz9 | 4cz9 |
| Model resolution (Å) | 3.69 | 3.6 |
| FSC threshold | 0.50 | 0.50 |
| Map sharpening B factor (Å²) | −77.5 | −133.8 |
| Model composition | | |
| Non-hydrogen atoms | 6131 | 6115 |
| Protein residues | 771 | 769 |
| Ligands | – | – |
| B factors (Å²) | | |
| Protein | 104.99 | 116.89 |
| Ligand | – | – |
| R.m.s. deviations | | |
| Bond lengths (Å) | 0.004 | 0.006 |
| Bond angles (°) | 0.525 | 0.783 |
| Validation | | |
| MolProbity score | 2.03 | 1.86 |
| Clashscore | 8.76 | 8.31 |
| Poor rotamers (%) | 1.5% | 0.5% |
| Ramachandran plot | | |
| Favoured (%) | 94.38 | 93.94 |
| Allowed (%) | 5.62 | 6.06 |
| Disallowed (%) | 0 | 0 |

[§]Correction added on 15 December 2020, after first online publication: The PDB codes for NHE9* and NHE9 ΔCTD were interchanged. The numbers 6131 and 6115 were moved from 'Model composition' to 'Non-hydrogen atoms'. The number of protein residues was changed from - to 771 and from - to 769 for NHE9* and NHE9 ΔCTD, respectively. The percentages of poor rotamers for NHE9* and NHE9 ΔCTD were interchanged.

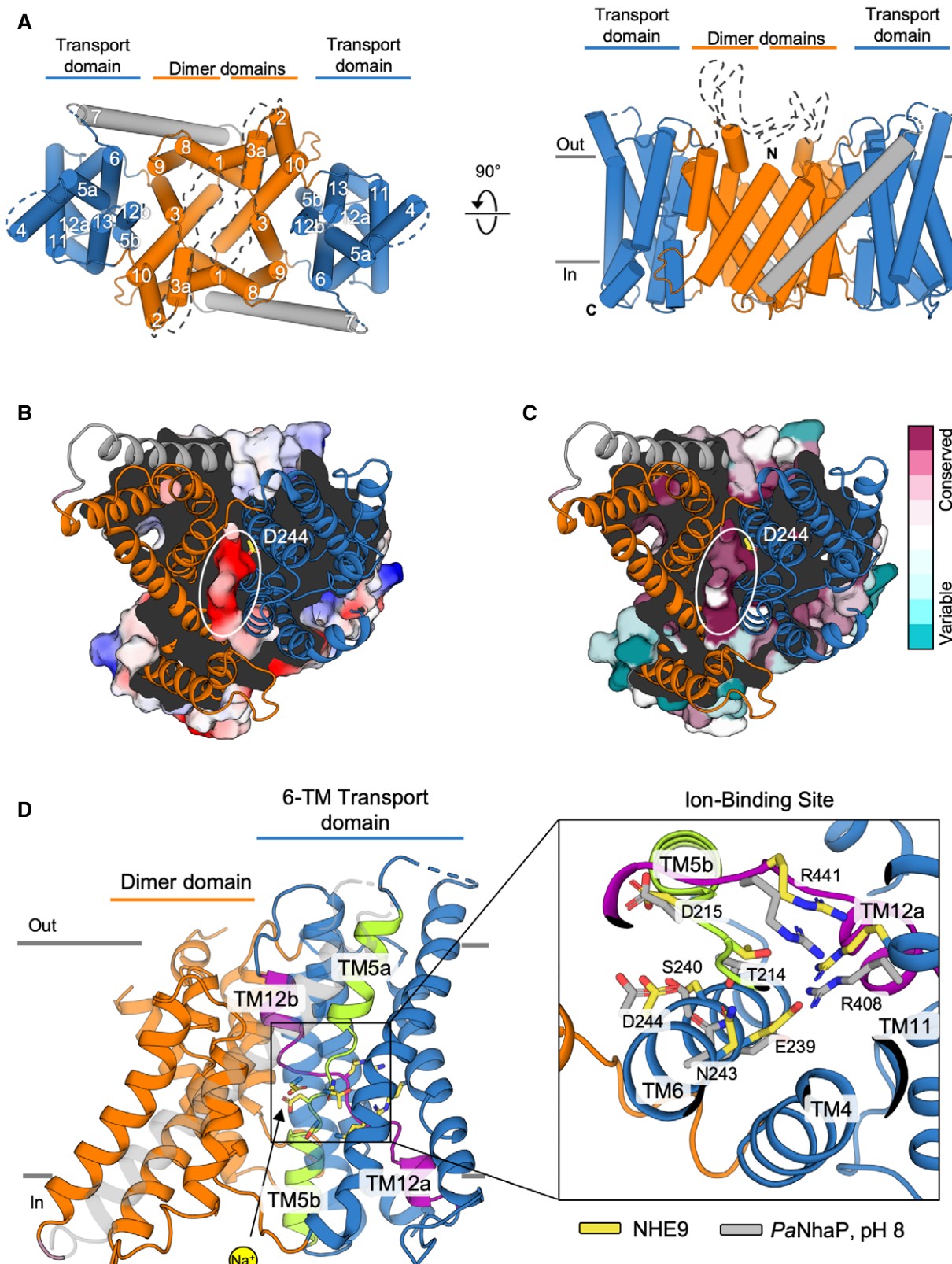

**Figure 2.**

◄

**Figure 2.  NHE9 architecture and ion-binding site of the inward-facing NHE9 homodimer.**

A   Cartoon representation of the NHE9 homodimer shown from the extracellular side in the endosomal lumen (left) and along the membrane plane (right). Ion translocation 6-TM core transport domains (blue), dimerization domains (orange) and linker helix TM7 (grey), are coloured as in Fig 1E with the respective helices numbered.

B   Cartoon representation of dimeric NHE9 ΔCTD from the cytosolic side with the electrostatic surface representation through the ion-binding site of one monomer (coloured blue to red, for positive to negative charges, respectively). The strictly conserved ion-binding residue Asp244 is labelled and shown in yellow as sticks. The inward open ion-binding cavity is encircled.

C   NHE9 ΔCTD as in (B), but coloured according to conservation scores from the alignment of 650 mammalian NHE1-9 representative sequences calculated with ConSurf server (Ashkenazy *et al*, 2016; see Materials and Methods).

D   *left*: cartoon representation of the NHE9 ion-binding site in the 6-TM core transport domain, which is made up of two broken helices TM5a-b (green) and TM12a-b (purple) accessible to a sodium ion from the cytoplasm (yellow sphere). *right*: ion-binding site residues are shown as yellow sticks and labelled with the corresponding residues in *Pa*NhaP (PDB id: 4cz8) shown as grey sticks.

potential (Ψ) (Materials and Methods and Fig 1C). In this experimental setup, proton efflux by NHE9* was ~ 3-fold higher than a NHE9* variant in which the critical ion-binding aspartic acid (D244) and the preceding asparagine (N243)—the strictly conserved "ND" motif (Brett *et al*, 2005; Masrati *et al*, 2018)—were substituted to alanine (Fig 1C and D). We could confirm the dequenching observed by the N244A-N243A variant was artefactual, since the unrelated fructose transporter GLUT5 gives a similar response (Fig 1C and D). Whilst an improved assay is required before one can make detailed comparisons between NHE9* variants, the activity was significant enough for the determination of an apparent $K_M$ of NHE9* for Na$^+$ (20.5 ± 2.9 mM), which was similar to estimates of endosomal isoforms NHE6 (10 mM) and NHE8 (23 mM) (Xu *et al*, 2005; Pedersen & Counillon, 2019; Fig 1D).

NHE9* sample preparation was optimized for grid preparation, cryo-EM data acquisition and structural determination at an active pH of 7.5 (Materials and Methods). We combined 3D classes from two independent data collections of ~ 1.4 million particles, from which an EM map was reconstructed to 3.5 Å according to the gold-standard Fourier shell correlation (FSC) 0.143 criterion, which used ~ 5% of the total collected particles (Table 1, Appendix Figs S2 and S3A). The EM maps were well resolved for the TMs, but a few of the connecting loop residues and a larger, extracellular loop of 51 residues, located between TM2 and TM3, could not be built (Fig EV2A). Surprisingly, there was no obvious density for any of the 93-residue long C-terminal tail regulatory domain (CTD), likely a result of its predicted dynamics and intrinsic disorder (Fig EV2A; Norholm *et al*, 2011; Hendus-Altenburger *et al*, 2014; Pedersen &

Counillon, 2019). We repeated cryo-EM structural determination for a *horse* NHE9 construct lacking the entire CTD domain (NHE9 ΔCTD), which displayed similar kinetics as the close-to-full-length NHE9* construct (Figs 1D and EV3 and Materials and Methods). Consistent with removal of the entire flexible CTD, 36% of the auto-picked particles now contributed to the final 3D class with an improved EM map resolution of 3.2 Å (Figs 1E, and EV2B and EV3, and Appendix Fig S3B). As model building revealed only minor differences between NHE9* and NHE9 ΔCTD structures, the later was used for all subsequent analysis due to its moderately better resolution (Fig EV2C).

The NHE9 monomer consists of 13 TMs with an extracellular N-terminus and intracellular C-terminus (Fig 2A and Appendix Fig S4A). The NHE9 structure is therefore more similar to the bacterial homologue structures with 13 TMs and a 6-TM topology inverted repeat, namely NapA (Lee *et al*, 2013), *Mj*NhaP (Paulino *et al*, 2014) and *Pa*NhaP (Wöhlert *et al*, 2014), rather than the more commonly used NHE models that, like NhaA, have 12 TMs and a 5-TM topology inverted repeat (Hunte *et al*, 2005; Landau *et al*, 2007; Kondapalli *et al*, 2013; Hendus-Altenburger *et al*, 2014; Pedersen & Counillon, 2019; Li *et al*, 2020; Appendix Fig S4B). The expansion of the inverted-topology repeats establishes a dimerization interface that in NHE9, and the bacterial antiporters with 13 TMs, is formed predominantly by tight interactions between TM1 on one monomer and TM8 on the other, burying a total surface area of ~ 1,700 to 2,000 Å$^2$ (Fig EV4A). In NhaA, with 12 TMs, the dimerization interface is instead formed by interactions between an antiparallel β-hairpin extension in a loop domain that buries a total surface area of only ~ 700 Å$^2$ (Lee *et al*, 2014). In NHE9, the substrate-binding

►

**Figure 3.  MD simulations of NHE9.**

A   Na$^+$ density from MD simulation (m2-00-f-3), measured in mol/l. The bulk density is ~ 150 mM. Only one binding site is visible with this cut through the density. The membrane is omitted for clarity.

B   Top view of the binding site; Na$^+$ positions are shown as small spheres, drawn at 10-ns intervals. Yellow/cyan/solid grey: simulation m2-00-p-2, which started with a Na$^+$ modelled near D244. Red/transparent red/transparent grey: simulation m2-00-f-3 during which a Na$^+$ ion spontaneously entered the binding site and bound in almost the exact same binding pose as in the simulation where the ion was modelled.

C   Cut through the density (same view as (B)) for simulation m2-00-f-3.

D   Cut through the density (same view as (B)) for the simulation m2-00-p-2 (over the part of the trajectory with a Na$^+$ bound).

E   Na$^+$ coordinating residues as see in MD (from trajectory m2-00-p-2, promoter A). The side-chain carboxylates of D244 and D215 interacted with the ion together with the backbone carbonyl oxygen for T214 and S240 and typically two water molecules.

F   Na$^+$ coordinating residues (from trajectory m2-00-p-2, promoter B). Only the side-chain carboxylate of D244 contributed to binding together with the backbone carbonyl oxygen of T214 and S240 and typically two water molecules.

G   Membrane lipid density from CG MD simulations. The combined POPC and cholesterol lipid density was contoured at the bulk density to show the extent of interactions between NHE9 (shown as tubes) and lipids. (left) Top view (from extracellular side) (right). Side (in-membrane) view. The circular shape of the membrane patch is an artefact from the rotational superposition of the orthorhombic simulation system prior to density analysis and indicates that the protein was rotating freely in the membrane.

					

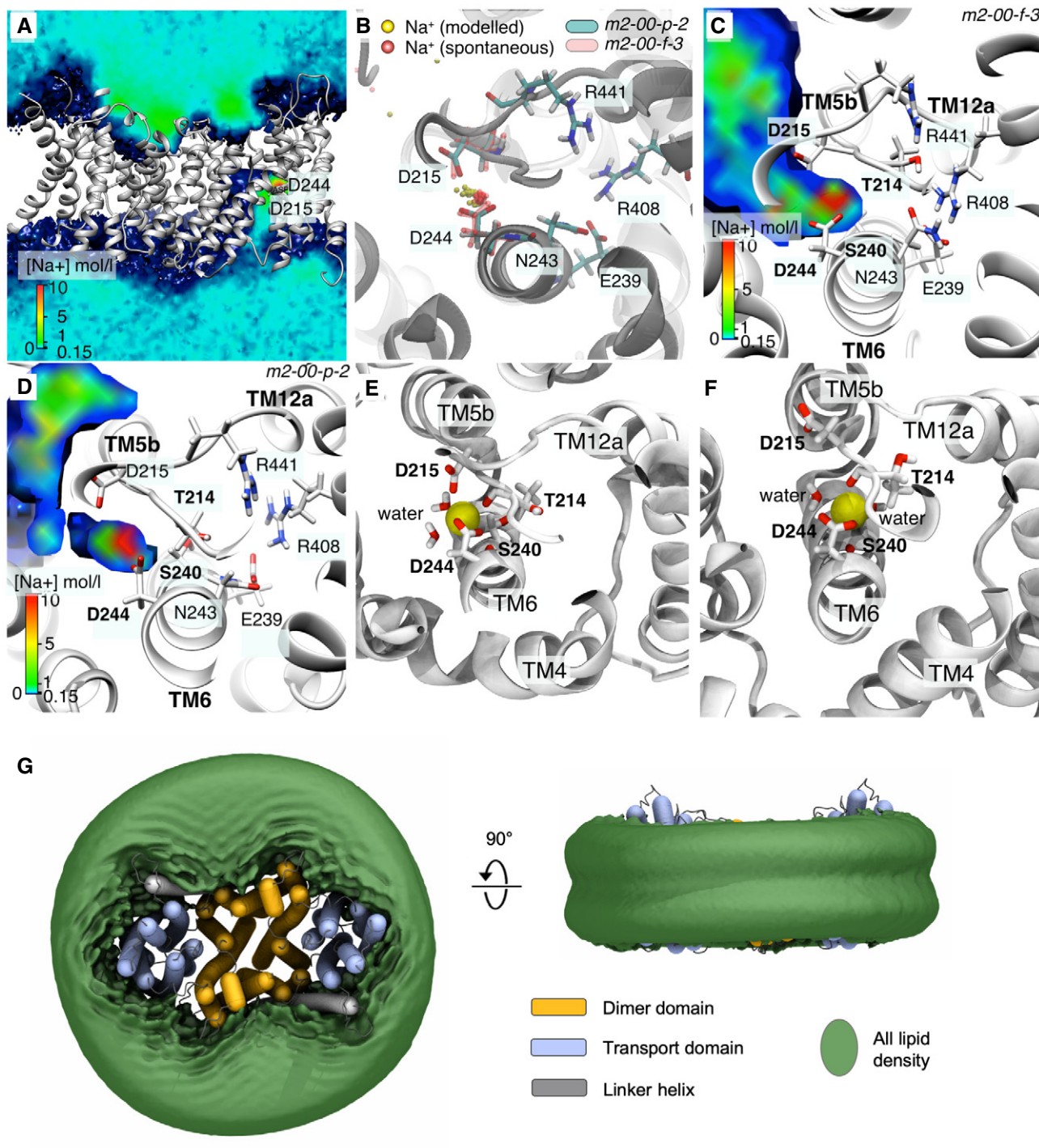

**Figure 3.**

cavity is located between the dimer and core ion-transport domains and is open towards the intracellular side (Figs 2B and EV4A). Near the base of the cavity is the strictly conserved aspartate Asp244 (TM6) (Figs 2B–D and EV1), essential for ion-binding and transport (Padan, 2008; Maes *et al*, 2012; Lee *et al*, 2013; Wöhlert *et al*, 2014; Coincon *et al*, 2016; Pedersen & Counillon, 2019). The ion-binding site and the negatively charged funnel are highly conserved across all NHEs (Figs 2B and C and EV1), enabling the generation of plausible models for all NHEs, e.g. including the clinical drug targets NHE1 and NHE3 (Pedersen & Counillon, 2019; Fig EV4B and Appendix Fig S5).

The 6-TM core domain is typified by two discontinuous helices TM5a-b and TM12a-b that contain highly conserved unwound regions that cross over each other near the centre of the membrane

(Figs 2A, and EV1 and EV5A). These extended helix break points harbour the strictly conserved residues Asp215 and Arg441, which are well-orientated to neutralize the ends of the oppositely charged half-helical TM5a and TM12b dipoles (Fig 2D). Asp215 is positioned at the cytoplasmic funnel and located opposite to Asp244 to create a negatively charged binding pocket readily accessible for a

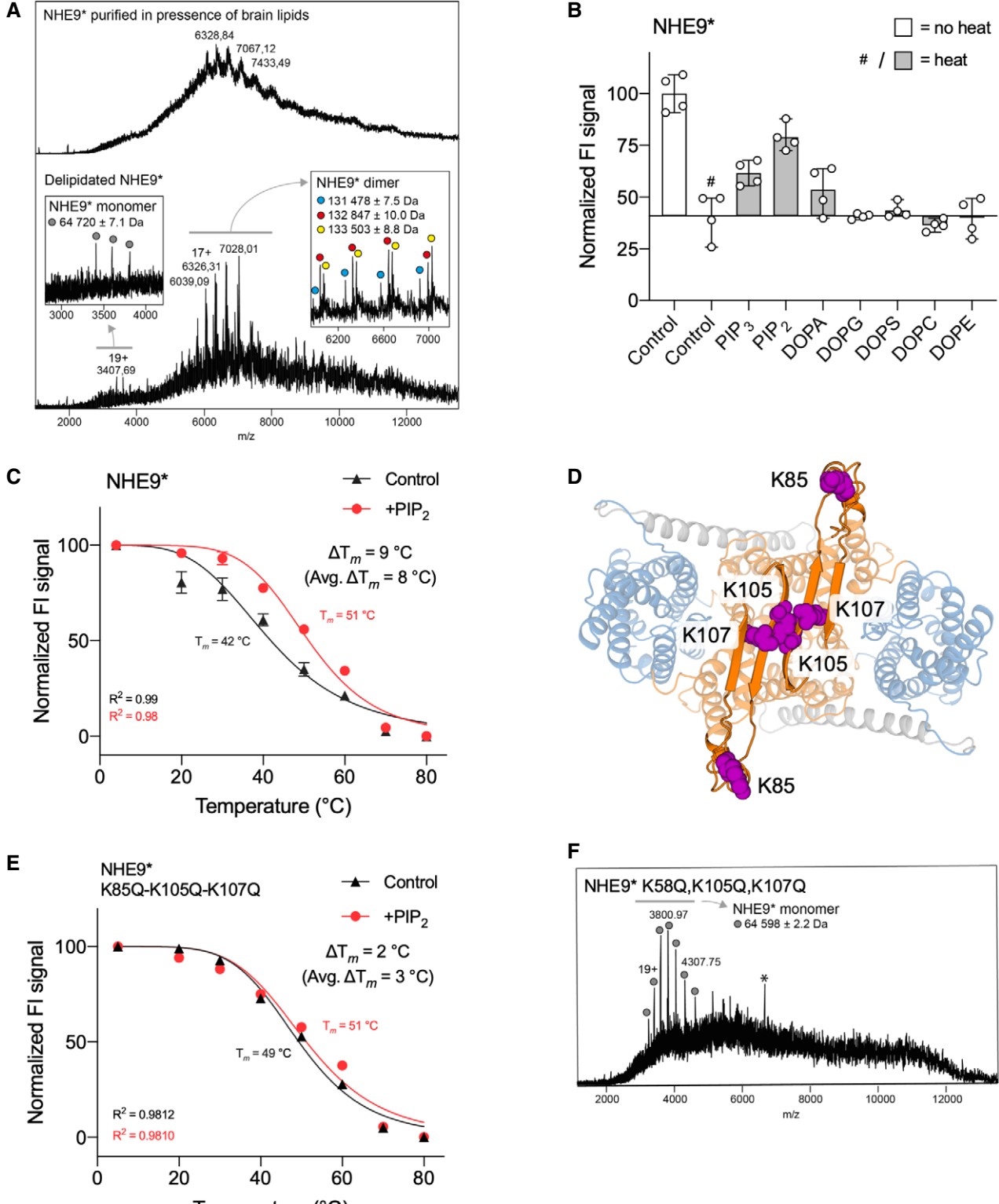

**Figure 4.**

◀

**Figure 4. Phosphoinositide-mediated stabilization of NHE9 dimer.**

A   Native mass spectrum of NHE9* purified in the presence (top panel) or the absence of additional brain lipids (bottom panel). In the presence of brain lipids, NHE9* is detected as exclusively dimeric protein with poorly resolved peaks, indicating the presence of multiple lipid adducts. Without brain lipid addition, NHE9* appears as sharp peaks, revealing several adducts of approximately 1 kDa each, as well as a minor monomer population.

B   Thermal stabilization of purified dimeric NHE9*-GFP by lipids. Normalized mean FSEC peak fluorescence before heating (open bars), and after heating and centrifugation (grey bars) in the presence of different lipids. Data presented are mean values ± data range of $n = 4$ experiments (see Materials and Methods).

C   Thermal shift stabilization of purified dimeric NHE9*-GFP in the presence of $PIP_2$ (red) compared to $PIP_2$-free (black). Data presented are normalized mean FSEC peak fluorescence as mean values ± data range of $n = 2$ technical repeats; the apparent $T_m$ was calculated with a sigmoidal 4-parameter logistic regression function; the average $\Delta T_m$ presented is calculated from $n = 2$ independent titrations.

D   Cartoon representation of NHE9 including a model of the loop between TM2-TM3, coloured as in Fig 2A. The positively charged residues Lys85, Lys105 and Lys107 are shown as purple spheres.

E   Thermal shift stabilization of the dimeric NHE9*-K85Q-K105Q-K107Q-GFP by $PIP_2$ (red) compared with thermal shift in the absence (black) of $PIP_2$; data shown are mean values ± data range of $n = 2$ technical repeats; the apparent $T_m$ was calculated with a sigmoidal 4-parameter logistic regression function; the average $\Delta T_m$ presented is calculated from $n = 2$ independent titrations.

F   Native MS of the triple mutant reveals monomeric NHE9 with no notable lipid adducts; a peak highlighted by * was determined to be a soluble contaminant as it was also apparent at MS conditions where no NHE9 was retained.

monovalent cation (Fig 2D). In MD simulations, $Na^+$ spontaneously bound to Asp215 and Asp244 residues, but no binding was observed when Asp244 was protonated (Appendix Table S1, Fig 3A and B, and Appendix Fig S6). Although the 6-TM core domain structure is highly conserved (Appendix Fig S4B), the ion-binding site residues themselves are most similar to PaNhaP and MjNhaP which, like NHE9, also catalyse electroneutral exchange (Figs 2D and EV5B). Specifically, in the electrogenic NhaA and NapA antiporters, the residue corresponding to Asn243 in TM6 of NHE9 is replaced by an aspartic acid that is further salt-bridged to a lysine residue in TM11; an interaction critical for electrogenic transport (Fig EV5C; Lee et al, 2014; Uzdavinys et al, 2017; Masrati et al, 2018). However, in NHE9 and other electroneutral exchangers, a parallel salt bridge is instead formed between Glu239 in TM6 and Arg408 in TM11 (Figs 2D and EV5C); as yet, the functional role for this highly conserved electroneutral salt bridge is unclear (Masrati et al, 2018).

The most noticeable difference between NHE9 and the bacterial electroneutral $Na^+/H^+$ antiporters is that they have additional negatively charged residues orientated towards the ion-binding site (Fig EV5B). In PaNhaP, one of these glutamate residues was observed coordinating a bound $Tl^+$ (Okazaki et al, 2019; Fig EV5B), but was not essential for function (Wöhlert et al, 2014). The $Tl^+$ site in PaNhaP might represent the initial binding site for the ion, whereas the transported $Na^+$ site of NHE9 could be positioned deeper into the core domain, as seen in MD simulations of NHE9 when both Asp244 and Asp215 were deprotonated. Based on the simulations, the $Na^+$ ion interacts with Asp244 and with the backbone carbonyl oxygen atoms of the polar residues Thr214 and Ser240, as well as water molecules (Fig 3C–F); Asp215 does not always directly coordinate the ion but needed to be deprotonated for binding to be observable.

NHE9 is highly expressed in the pre-frontal cortex of the brain (Zhang-James et al, 2019). The Slc9a9 knockout mouse and rat models exhibit autism spectrum disorders-like behavioural deficits (Yang et al, 2016). NHE9 variants Val177Leu, Leu236Ser, Ala409Pro, Ser438Pro and Arg423X identified in patients with autism and/or epilepsy (Zhang-James et al, 2019) cluster within the 6-TM core domain (Fig EV5D). The residues are located next to well-conserved glycine and proline residues that are typically required for protein dynamics, as was apparent by elastic network modelling (Appendix Fig S7A and B; Bahar et al, 2010). Thus, almost certainly the autism-associated mutations result in impaired

NHE9 activity, consistent with the initial evaluation of these variants in astrocytes (Kondapalli et al, 2013).

The membrane composition is critical for NHE activation and associated with NHE1 mechanosensation and cell volume regulation (Fuster et al, 2004; Pedersen & Counillon, 2019). In NHE9, the end of TM3 was observed to have moved away from the dimerization interface, exposing a hydrophobic surface that could bind lipids between the protomers, reminiscent of the lipid binding seen in PaNhaP at active pH and in the distantly related citrate transporter CitS (Wöhlert et al, 2014; Xu et al, 2020; Fig EV4A). Interestingly, 3D variability analysis (Punjani & Fleet, 2020) of the NHE9 ΔCTD reconstruction reveals several conformational states in which the protomers "breathe" further apart at the dimerization interface (Movie EV1 and Appendix Fig S7C). In coarse-grained MD simulations, NHE9 is stably embedded in the membrane (7:3 POPC : cholesterol) throughout a total of ~ 400 μs, and only the extracellular ends of the core domain helices extended above the average surface of the lipid environment (Fig 3G). Although lipids might bind at the dimerization interface to stabilize these motions, in MD simulations of the NHE9 structure, lipids were unable to diffuse into the interface (Fig 3G).

To assess lipid preferences in detergent solution, purified NHE9* was analysed by native mass spectrometry (MS) (Materials and Methods). When brain lipids were added during purification, NHE9* formed dimers and multiple lipid adducts were observed (Materials and Methods and Fig 4A). However, without addition of brain lipids both NHE9* monomer and dimers were detected, and only the NHE9* dimers were observed with lipid adducts of ~ 1 kDa in mass. However, it was not possible to resolve the exact lipid identity by native MS. We therefore screened lipid preferences of NHE9* using a GFP-based thermal shift assay, which we had previously shown was able to identify the specific lipid-binding preference of NhaA, that of cardiolipin (Gupta et al, 2017; Nji et al, 2018). Amongst a number of functionally relevant lipids, clear thermostabilization of NHE9* was only apparent after addition of the negatively charged lipids $PIP_2$, and $PIP_3$ (Fig 4B). In the presence of either $PIP_2$ or $PIP_3$, the average melting temperature ($\Delta T_m$) of NHE9* increased by 9°C—in comparison, POPC, POPE and POPA lipids showed no clear stabilization (Fig 4C, Appendix Figs S8A and S9). Thus, we could confirm an interaction to lipids with masses consistent to the additional adducts detected by native MS in the NHE9 dimer.

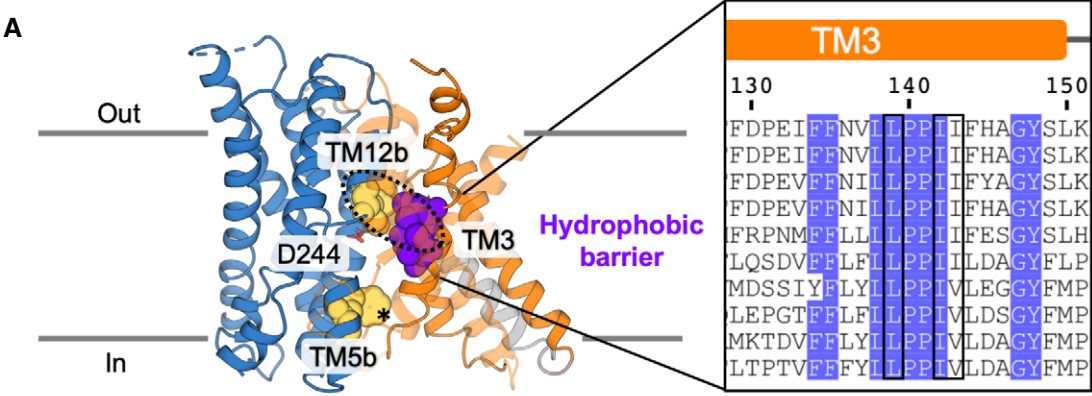

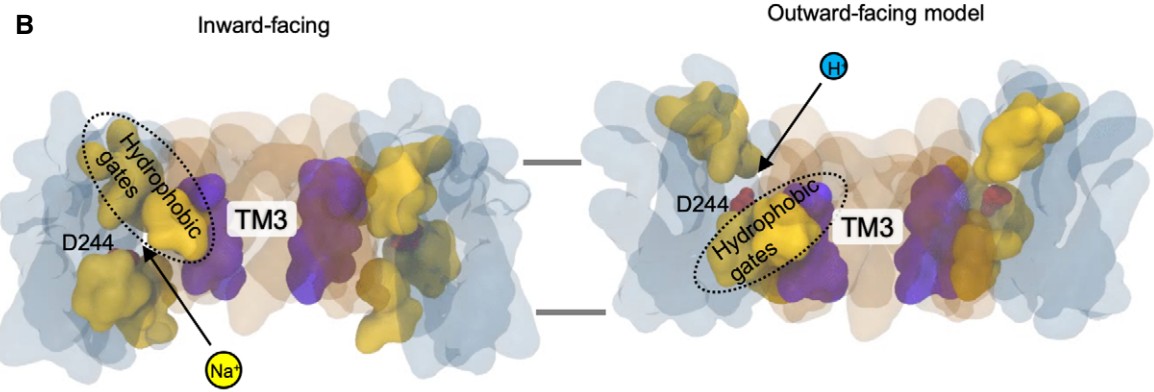

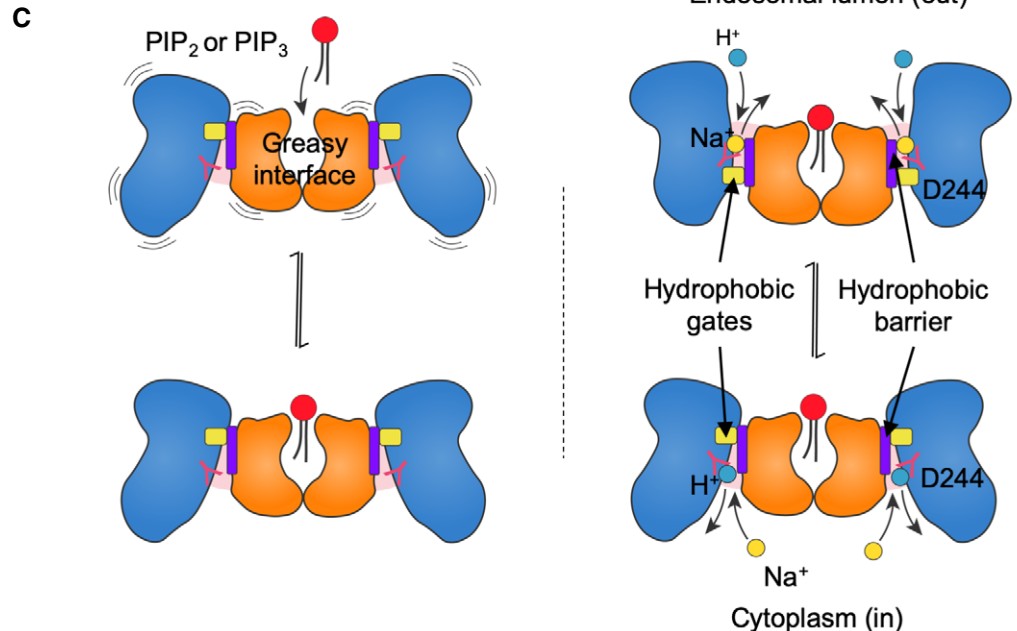

**Figure 5.**

**Figure 5. The NHE9 elevator alternating access mechanism.**

A   Side view of NHE9 monomer constituted by 6-TM core transport domain (blue), dimerization domain (orange) and linker helix TM7 (grey) showing the extracellular hydrophobic gate in the inward conformation between TM12b (yellow spheres) and TM3 (purple spheres and black boxes in alignment). The hydrophobic barrier is made up of highly conserved residues in TM3 of the dimer domain that face towards the transport domain; indicated as purple spheres. The asterisk (*) indicates hydrophobic residues in TM5b of the transport domain that are likely to form an intracellular hydrophobic gate in the outward-facing state as based on outward NapA.

B   Inward-facing (left) and outward-facing (right) NHE9 model snapshots along a trajectory following the major principal component (PC1) derived from analysis of the conserved Na$^+$/H$^+$ antiporter structural alignment (see Materials and Methods). Extracellular and intracellular hydrophobic gates are coloured as in (A).

C   Schematic representation of NHE elevator alternating access mechanism highlighting the role of negatively charged phosphoinositide lipids to stabilize homodimerization (left) and the hydrophobic gates formed either side of the ion-binding aspartate (right).

PIP$_2$ has been demonstrated to be essential for NHE1 function (Aharonovitz *et al*, 2000; Abu Jawdeh *et al*, 2011; Pedersen & Counillon, 2019), and NHE3 can be directly and reversibly activated in seconds by PIP$_3$ (Abu Jawdeh *et al*, 2011). In NHE1, this occurs in part due to direct interactions of PIP$_2$ to the CTD (Shimada-Shimizu *et al*, 2014; Pedersen & Counillon, 2019). To investigate this possibility in NHE9, the ΔCTD construct was also analysed, yet additional stabilization upon PIP$_2$ addition was still observed (Appendix Fig S8B). Since PIP$_2$ binding correlated with dimer stability during native MS, the lipid-binding site should presumably be located at the dimerization interface, yet no positively charged residues—required for phosphoinositide coordination (McLaughlin *et al*, 2002)—were observed in our cryo-EM model. We noticed that the unmodelled TM2-TM3 loop domain contained three lysine residues (Fig EV1). In the cryo-EM maps, density for the loop domain was observed above the dimer interface (Fig EV2A). The TM2–TM3 loop is predicted to form β-strands sharing homology to the β-strands in NhaA (Fig EV2A and Appendix Fig S4B). The TM2-TM3 loop in NHE9 is situated in the same position as the β-hairpins in NhaA, which mediate dimerization (Lee *et al*, 2014). Moreover, the negatively charged cardiolipin is thought to bind at the same interface to further stabilize the NhaA homodimer (Gupta *et al*, 2017; Landreh *et al*, 2017; Nji *et al*, 2018; Appendix Fig S8C and Fig 4D). Consistently, in NHE9 a triple lysine-to-glutamine loop mutant markedly reduced PIP$_2$-mediated stabilization, and only delipidated monomers were observed by native MS (Fig 4E and F). Taken together, the dimer interface in NHE9 is dynamic and phosphoinositides can bind to the TM2-TM3 loop domain to stabilize the functional dimer, analogous to how negatively charged cardiolipin stabilizes the NhaA dimer in *E. coli* (Gupta *et al*, 2017; Nji *et al*, 2018; Rimon *et al*, 2019).

Overall, the dimerization domain is structurally more divergent than the 6-TM core domain (Fig EV1 and Appendix Fig S4B). The only highly conserved sequence in the dimer domain is located on TM3, with the consensus "FFXXLLPPI[IV]" (Figs EV1 and 5A and Appendix Fig S10A). The TM3 residues in the dimer domain create a hydrophobic surface, which in NHE9 is closed-off above the ion-binding site by hydrophobic interactions with the 6-TM core domain (Fig 5A and Appendix Fig S10B). The hydrophobic contacts between Val366 and Ile248 residues in NHE9 are, in fact, equivalent to the extracellular "hydrophobic gate" recently identified from MD simulations of *Pa*NhaP (Okazaki *et al*, 2019) and partly overlap with the phenylalanine gates firstly noted in NhaA (Hendus-Altenburger *et al*, 2014). Bioinformatic contact analysis further uncovers highly conserved hydrophobic residues that are also located between the core and dimer domains in the outward-facing state of NapA (Fig 5A and Appendix Fig S10B). Thus, there are evolutionarily

conserved hydrophobic gates, either side of the ion-binding site, that form a seal between the 6-TM core and dimer domains presumably to prevent ion leakage.

Given the high degree of structural similarity between NHE9 and the bacterial homologous structures, we can construct a reliable structural alignment with a r.m.s.d 3.0 ± 1.3 Å (Materials and Methods and Appendix Fig S11). Although NHE9 is an inward-facing conformation, principal component analysis (PCA) places it closer to the outward-facing state of NapA than previously determined inward-facing antiporter structures (Appendix Fig S11). This structural classification is most noticeable in the electrostatic surface representation, as the ion-binding site of NHE9 is located further from the cytoplasmic surface compared with the closest structural homologue *Pa*NhaP and, consequently, the barrier to the outside is also thinner (Fig EV4A). Indeed, the 6-TM core domain in NHE9 is more detached from the dimerization domain surface sharing a buried surface area of only 960 Å$^2$ as compared to 1,500 Å$^2$ in *Pa*NhaP (Fig EV4A and Appendix Table S2). It is possible that the inward NHE9 structure represents an "unlocked" state as seen in other elevator proteins (Akyuz *et al*, 2015; Drew & Boudker, 2016). Remarkably, elastic network modelling of NHE9 reveals intrinsic dynamics that covers ~ 80% of the transition to the outward state in NapA (Movie EV2 and Appendix Fig S7A). Structural transitions between the here presented inward-facing state of NHE9 to the outward state in NapA would require a vertical displacement of the ion-binding site by up to 5 Å, which is consistent with the degree of vertical movement modelled by MD-based transition-path sampling of *Pa*NhaP (Okazaki *et al*, 2019; Fig 5B, Appendix Fig S11B and Movie EV3).

## Discussion

The highly-conserved structure of the ion-binding site shared between NHE9 and the bacterial Na$^+$/H$^+$ antiporters, particularly Tl$^+$-bound *Pa*NhaP, highlights that ion-coordination, hydrophobic gating and structural transitions are all likely to be similar, as supported here by computational analysis. Nonetheless, with the differences in ionisable side-chain interactions to the conserved ion-binding residues in NHE9 and the bacterial homologues (Fig EV5B), it seems the ion-binding site has been fine-tuned differently, perhaps to match individual NHE function and energetics. Furthermore, we expect that there is further ion induced structural rearrangements yet to be realised in the absence of an ion-bound intermediate. For example, it was shown that the ion-binding aspartate rotates from the cavity towards the 6-TM core domain when transitioning from the outward- to the inward-facing conformation in NapA (Coincon

*et al*, 2016); a local rearrangement also observed in MD simulations of *Pa*NhaP (Okazaki *et al*, 2019).

Given that pH varies greatly amongst different organelles, it is unsurprising that intracellular pH and NHE activities are stringently regulated (Casey *et al*, 2010). Our analysis demonstrates that stability of the NHE9 homodimer is associated with the binding of the negatively charged lipids $PIP_2$ and $PIP_3$ (Fig 5C). Since phosphoinositides are intimately involved in most, if not all, vesicular transport pathways (Mayinger, 2012), it is tempting to speculate that *in vivo* phosphoinositide lipids allosterically regulate NHE9 activity. This hypothesis stems from the fact that (i) phosphoinositide lipids are well known to activate NHE1 and NHE3 (Pedersen & Counillon, 2019) and (ii) the negatively charged lipid cardiolipin is needed for optimal NhaA function in *E. coli* (Padan, 2008; Rimon *et al*, 2019); indeed, cardiolipin synthesis is upregulated by salt stress and NhaA is required to export excess $Na^+$ under salt-stress conditions (Romantsov *et al*, 2009). Furthermore, to the best of our knowledge, the NhaA fold is the only transporter fold seen to date with an expansion in the structural-inverted repeat (Drew & Boudker, 2016). Such variation, establishes a different dimerization interface, which further implies that there has been evolutionary pressure to modulate oligomerization. Our study therefore contributes to the growing body of evidence that secondary-active transporters, especially elevator proteins, might be regulated by controlling when and how they oligomerize (Nji *et al*, 2018; Cecchetti *et al*, 2019; Diallinas & Martzoukou, 2019).

The large and varied cytosolic domain is unique to the eukaryotic NHE members, and is known to interact with many different partner proteins, lipids and hormones to control their ion-exchange activity, trafficking and downstream signalling (Pedersen & Counillon, 2019). The CTD in NHE1, for example, interacts with $Ca^{2+}$ calmodulin (CaM), which has been proposed to activate NHE1 by re-modelling the tail so that it no longer interacts with the transporter module in an inhibitory manner (Wakabayashi *et al*, 1997; Pedersen & Counillon, 2019). Despite extensive efforts, we were unable to obtain any structural information for the regulatory domain of NHE9. It, therefore, seems unlikely that the C-terminal tail in NHE9 interacts tightly with the transporter module in the absence of binding partners. Nevertheless, since the regulatory domain in NHE9 lacks the $Ca^{2+}$-CaM-binding motifs (Bertrand *et al*, 1994), we cannot assess if this auto-inhibitory mechanism occurs in NHE1 and other isoforms harbouring $Ca^{2+}$-CaM-binding sites. In addition to CTD, the length of the N-terminus also varies, from 23 residues upstream of TM1 in NHE9, to 84 amino acids upstream of the modelled TM1 in NHE1 (Fig EV1). In NHE1, the N-terminal region is quite charged and glycosylated (Pedersen & Counillon, 2019), but its removal by proteolysis does not affect function (Shrode *et al*, 1998). The N-terminal region upstream of TM1 in all NHE isoforms —apart from NHE8—harbours a predicted signal peptide (Almagro Armenteros *et al*, 2019; Fig EV1), as noted previously (Landau *et al*, 2007). Though most topology models of NHE1 have interpreted this region as the first TM segment (Pedersen & Counillon, 2019), based on the NHE9 structure and from a biogenesis perspective, it makes most sense as a cleavable signal peptide, since its presence correlates best with the length of the N-terminus, i.e. large and charged extracellular N-terminal tails often have a signal peptide, as they cannot otherwise be efficiently translocated by Sec61 (Wallin & von Heijne, 1995).

To conclude, the NHE9 structure provide a rational explanation as to why autism-related disease mutants disrupt NHE9 activity, and provides a suitable template for homology modelling other clinically-relevant NHE isoforms, such as NHE1 and NHE3. Based on NHE9 intrinsic dynamics and overall structural conservation with bacterial antiporters, our analysis further supports that alternating-access to the ion-binding site is achieved by elevator-like structural transitions (Fig 5C). This study therefore enhances our mechanistic understanding of mammalian NHEs and brings us closer to their pharmacological control. Nonetheless, much work remains before we can discern the mechanistic basis for how NHEs are allosterically regulated by their C-terminal domains, which is critical step for full interpretation of their physiological functions.

# Materials and Methods

## Target identification using fluorescence-based screening methods

NHE9 genes from *Homo sapiens*, *Mus musculus*, *Bos taurus*, *Rattus norvegicus*, *Equus caballus*, *Oryctolagus cuniculus*, *Canis lupus familiaris*, *Heterocephalus glaber*, *Aptenodytes forsteri*, *Gallus gallus*, *Columba livia*, *Anas platyrhynchos* and *Corvus brachyrhynchos* were synthesized and cloned into the GAL1-inducible TEV site containing GFP-TwinStrep-His₈ vector pDDGFP3, and transformed into the *S. cerevisiae* strain FGY217 (MATα, ura3–52, lys2Δ201 and pep4Δ) as previously described (Kota *et al*, 2007). The cloned constructs were subsequently screened for recombinant protein production and stability as described previously (Drew *et al*, 2008). In brief, the respective cloned construct was first produced in 50-ml bioreactor tubes and screened for protein production levels by detecting the total GFP fluorescence of harvested cells in a 96-well plate reader (485 $nm_{ex}$, 538 $nm_{em}$). Following, the highest yielding construct was overexpressed in 2 l cultures, cells harvested, and membranes isolated and solubilized with 1% (w/v) n-dodecyl β-D-maltoside (DDM, Glycon). Subsequently, the mono-dispersity of the detergent-solubilized protein product was assessed using fluorescence-detection size-exclusion chromatography (FSEC) using a Shimadzu HPLC LC-20AD/RF-20A (488 $nm_{ex}$, 512 $nm_{em}$) instrument and Superose 6 10/300 column (GE Healthcare) in 20 mM Tris–HCl, pH 7.5, 150 mM NaCl, 0.03% (w/v) DDM. The thermostability of the highest expressing and most mono-disperse candidate constructs was determined as described previously (Kawate & Gouaux, 2006; Drew *et al*, 2008). Of the NHE9 homologues screened, *horse* NHE9 (UniProt accession: F7B133) showed the sharpest mono-disperse FSEC peak and the highest thermostability after purification. To further improve protein yield and stability, the C-terminal tail of NHE9 was partially truncated to generate NHE9* as it was predicted to be highly disordered by RONN (Yang *et al*, 2005).

The *horse* NHE9* construct contains 8–574 out of 644 residues; the additional C-terminal residues retained after TEV cleavage are shown in italics. Residues truncated in the NHE9 Δ CTD variant that contains 8–481 residues out of 644 are underlined. All *horse* NHE9 variants were generated with a standard PCR-based strategy as previously described (Drew *et al*, 2008).

MSEKDEYQFQHQGAVELLVFNFLLILTILTIWLFKNHR
FRFLHETGGAMVYGLMGLILRYATAPTDIESGTVYDCGKL

AFSPSTLLINITDQVYEYKYKREISQHNIPHLGNAILEKM
TFDPEIFFNVLLPPIIFHAGYSLKKRHFFQNLGSILTYAF
LGTAISCIVIGLIMYGFVKAMVYAGQLKNGDFHFTDCLFF
GSLMSATDPVTVLAIFHELHVDPDLYTLLFGESVLNDAVA
IVLTYSISIYSPKENPNAFDAAAFFQSVGNFGIFAGSFAM
GSAYAVVTALLTKFTKLCEFPMLETGLFFLLSWSAFLSAE
AAGLTGIVAVLFCGVTQAHYTYNNLSLDSKMRTKQLFEFM
NFLAENVIFCYMGLALFTFQNHIFNALFILGAFLAIFVAR
ACNIYPLSFLLNLGRKHKIPWNFQHMMMFSGLAIAFALAI
RDTESQPKQMMFSTTLLLVFFTVWVFGGGTT<u>PMLTWLQIR
VGVDLPSSHQEANNLEKSTTKTESAWLFRMWYGFDHKYLK
PILTHSGPPLTTTLPEWGPISRLLTSPQAYGEQLKE</u>*GEN
LYFQ*

## Large-scale protein production and purification of *horse* NHE9* and variants

*Saccharomyces cerevisiae* (FGY217) was transformed with the respective vector and cultivated in 24-l cultures—URA media at 30°C 150 RPM in Tuner shaker flasks using Innova 44R incubators (New Brunswick). At an $OD_{600}$ of 0.6 AU, protein overexpression was induced by the addition of galactose to final concentration of 2% (w/v) and incubation continued at 30°C, 150 RPM. Cells were harvested after 22 h by centrifugation (5,000 × g, 4°C, 10 min), resuspended in cell resuspension buffer (CRB, 50 mM Tris–HCl pH 7.6, 1 mM EDTA, 0.6 M sorbitol) and subsequently lysed by mechanical disruption as previously described (Drew *et al*, 2008). Cell debris was removed by centrifugation (10,000 × g, 4°C, 10 min), and from the resulting supernatant, membranes were subsequently isolated by ultracentrifugation (195,000 × g, 4°C, 2 h) and homogenized in membrane resuspension buffer (MRB 20 mM Tris–HCl pH 7.5, 0.3 M sucrose, 0.1 mM $CaCl_2$).

For samples used for structural studies by cryo-EM, membranes were solubilized in solubilization buffer (1% (w/v) lauryl maltose neopentyl glycol (LMNG, Anatrace), 0.2% (w/v) cholesteryl hemisuccinate (CHS, Sigma-Aldrich), 20 mM Tris–HCl pH 8.0, 150 mM NaCl, 10% (w/v) glycerol), during mild agitation for 1 h at 4°C and subsequently cleared by ultracentrifugation (195,000 × g, 4°C, 45 min). The resulting supernatant was incubated for 2 h at 4°C with 5 ml of Strep-Tactin XT resin (IBA Lifesciences) pre-equilibrated in wash buffer 1 (WB1, 0.1% (w/v) LMNG, 0.02% (w/v) CHS, 20 mM Tris–HCl pH 8.0, 150 mM NaCl). This resin was transferred into a gravity flow column (Bio-Rad) and subsequently washed in two steps, initially with 300 ml WB1 and finally with 600 ml wash buffer 2 (WB2, 0.003% (w/v) LMNG, 0.0006% (w/v) CHS, 20 mM Tris–HCl pH 8.0, 150 mM NaCl). Subsequently, the resin was resuspended in 20 ml of WB2 containing 0.003 mg/ml bovine brain extract lipids (Sigma-Aldrich, cat. nr. B3635) and incubated overnight at 4°C in the presence of equimolar amounts of TEV-His8 protease during mild agitation. The digested protein was collected, concentrated using 100 kDa MW cut-off spin concentrators (Amicon, Merck-Millipore) and subjected to size-exclusion chromatography (SEC), using a Superose 6 increase 10/300 column (GE Healthcare) and an Agilent LC-1220 system in 20 mM Tris–HCl pH 7.5, 150 mM NaCl, 0.003% (w/v) LMNG, 0.0006% (w/v) CHS.

For samples used for thermal stabilization studies or functional studies, the above-described protocol was used with the following alterations: (i) the solubilization buffer used was 20 mM Tris–HCl, pH 8.0 1% (w/v) DDM and 0.2% (w/v) CHS. (ii) Instead of digesting the protein by TEV and collecting the digested material, the protein was eluted in 20 mM Tris–HCl, pH 8.0 0.03% (w/v) DDM and 0.006% (w/v) CHS, 1 mM biotin. For functional studies, the above-described protocol was used with the following alteration; SEC was performed in 20 mM Tris–HCl, pH 8.0 0.03% (w/v) DDM and 0.006% (w/v) CHS.

## Native mass spectrometry

Purified NHE9* and variants were exchanged into 100 mM ammonium acetate, pH 7.0, containing 0.02% (w/v) DDM using ZebaSpin desalting columns (Thermo Scientific). The samples were introduced into the mass spectrometer using gold-coated borosilicate capillaries produced in-house. Mass spectra were recorded on hybrid Q-Exactive EMR mass spectrometer (Thermo Fisher) modified for high *m/z* analysis (Gault *et al*, 2016; Gupta *et al*, 2017). Instrument settings were as follows: capillary voltage, 1.4 kV; S-lens RF 100%; in-source trapping 300V; HCD collision energy 300 V; HCD cell pressure $1 \times 10^{-9}$ mbar; collision gas argon. Data were analysed using the Thermo Excalibur software package.

## Determination of the lipid preferences of NHE9* and variants by GFP-based thermal shift assay

To characterize the thermostability and lipid thermal stabilization of NHE9* and variants, we utilized the previously described FSEC-TS and GFP-TS assays (Hattori *et al*, 2012; Nji *et al*, 2018). In brief, GFP-fusions of purified NHE9* and variants were purified as described previously in buffer containing 0.03% DDM 0.006% CHS. Samples were diluted to a final concentration of 0.05–0.075 mg/ml and incubated in the presence of 1% (w/v) DDM for 30 min at 4°C. Subsequently, β-D-Octyl glucoside (Anatrace) was added to a final concentration of 1% (w/v) and the sample aliquots of 40 μl were heated for 10 min over a temperature range of 20–80°C in a PCR thermocycler (Veriti, Applied Biosystems) and heat-denatured material pelleted at 18 000 × g during 30 min at 4°C. The resulting supernatants were collected and analysed using FSEC (Shimadzu HPLC LC-20AD/RF-20A and Bio-Rad EnRich 650 column). The apparent $T_m$ was calculated by plotting the average GFP fluorescence intensity of the dimer of the respective NHE9* or variants from two technical repeats per temperature and fitting the curves to a sigmoidal 4-parameter logistic regression in GraphPad Prism software. Two technical repeats were considered sufficient for accurate $T_m$ calculations as the goodness of the fit was > 0.98 and the range between two technical repeats was low. The $\Delta T_m$ was calculated by subtracting the apparent $T_m$ in the presence and absence (control) of individual lipids. The presented $\Delta T_m$ reports the average from two independent measurements performed in two technical repeats.

Stock solutions of the respective lipids 1,2-dioleoyl-sn-glycero-3-phosphoethanolamine (DOPE, Avanti cat. no. 850725P), 1,2-dioleoyl-sn-glycero-3-phospho-(1′-rac-glycerol) (DOPG, Avanti cat. no. 840475P), 1,2-dioleoyl-sn-glycero-3-phosphocholine (DOPC, Avanti cat. no. 850375P), 1,2-dioleoyl-sn-glycero-3-phospho-L-serine (DOPS, Avanti cat. no. 840035P), 1,2-dioleoyl-sn-glycero-3-phosphoethanolamine (DOPE, Avanti cat. no. 840475P),

phosphatidylinositol tris-3,4,5-phosphate,1,2-dipalmitoyl (PIP$_3$, Larodan cat. no. 59-1102) and phosphatidylinositol bis-4,5-phosphate,1,2-dioctanoyl (PIP$_2$, Larodan cat. no. 59-1124) were prepared by solubilization in 10% ($w/v$) DDM to a final concentration of 10 mg/ml overnight at 4°C with mild agitation. To screen for differential lipid stabilization, the FSEC-TS protocol was utilized with the modification that the protein was incubated for 30 min at 4°C with the individual lipids at a final concentration of 1 mg/ml in 1% ($w/v$) DDM and these samples were heated at a single temperature ($T_m$ of the respective construct + 5°C). In all experiments, data were recorded from two biological repeats of two technical repeats and the plotted data error bars in GraphPad Prism show range of data.

### Coupled proton transport assay of NHE9* and variants

L-α-phosphatidylcholine lipids from soybean (soybean lipids, type II; Sigma-Aldrich, cat. nr. C6512) were solubilized, in 10 mM MES-Tris, pH 6.5, 5 mM MgCl$_2$, 100 mM KCl, to a final concentration of 10 mg/ml ($w/v$). The lipid solution was flash-frozen in liquid nitrogen and then thawed, throughout in a total of eight freeze–thaw cycles. The resulting liposomes were subsequently extruded (Lipos-Fast-Basic, Avestin) using polycarbonate filters with a pore size of 200 nm (Whatman). For protein reconstitution into liposomes, 250 μl of liposomes was destabilized by the addition of sodium cholate (0.65% ($w/v$) final concentration) and mixed with either 100 μg of NHE9* or variants or the unrelated fructose transporter rat GLUT5 as a negative control (Nomura *et al*, 2015) and *E. coli* F-ATPase (~ 1.5–3 mg/ml), and finally, the suspension was incubated for 30 min at RT. Excess detergent was removed using a PD-10 desalting column (GE), and the proteoliposomes were retrieved in a final volume of 2.3 ml. 100 μl of prepared proteoliposome solution was diluted into 1.5 ml of reaction buffer (10 mM MES-Tris, pH 8.0, 5 mM MgCl$_2$, 100 mM KCl) containing 2.7 μM 9-amino-6-chloro-2-methoxyacridine (ACMA, Invitrogen) and 130 nM valinomycin (Sigma-Aldrich). An outward-directed pH gradient (acidic inside) was established by the addition of ATP to a final concentration of 130 μM, and gradient formation was detected by a change in ACMA fluorescence at 480 nm using an excitation wavelength of 410 nm in a fluorescence spectrophotometer (Cary Eclipse; Agilent Technologies). After ~ 3-min equilibration, the desired concentration of NaCl was added and the transport activity of NHE9 was assessed by a change in ACMA fluorescence intensity (FI). The addition of NH$_4$Cl, to a final concentration of 20 mM was used to dissipate the proton motive force by quenching ΔpH. The percentage of ACMA dequenching was calculated as follows: $(FI_{4\ min} - FI_{3\ min})/(FI_{5\ min}/FI_{3\ min})*100$. For kinetic analysis, the response to increasing concentrations of NaCl was fitted from triplicate measurements to the Michaelis–Menten equation by nonlinear regression using the GraphPad Prism software. The final $K_M$ values reported are the mean ± s.d. of $n = 3$ independent (protein reconstitution) experiments.

### Cryo-EM sample preparation and data acquisition

Either 1.5 μg of purified NHE9* or 1.5 μg of purified NHE9 ΔCTD samples was individually applied to freshly glow-discharged Quantifoil R2/1 Cu300 mesh grids (Electron Microscopy Sciences). Grids were blotted for 3.0 s or 3.5 s under 100% humidity and plunge frozen in liquid ethane using a Vitrobot Mark IV (Thermo Fisher Scientific).

Cryo-EM datasets were collected on a Titan Krios G2 electron microscope operated at 300 kV equipped with a GIF (Gatan) and a K2 summit direct electron detector (Gatan) in counting mode. The movie stacks were collected at 165,000× corresponding to a pixel size of 0.83 Å at a dose rate of 7.0–8.0 e$^-$ per physical pixel per second. The total exposure time for each movie was 10 s, thus leading to a total accumulated dose of 80 e$^-$/Å$^2$, which was fractionated into 50 frames. All movies were recorded with a defocus range of −0.7 to −2.5 μm. The statistics of cryo-EM data acquisition are summarized in Table 1.

### Image processing

Dose-fractionated movies were corrected by using MotionCorr2 (Zheng *et al*, 2017). The dose-weighted micrographs were used for contrast transfer function estimation by CTFFIND-4.1.13 (Rohou & Grigorieff, 2015). The dose-weighted images were used for auto-picking, classification and reconstruction. For NHE9* and NHE9 ΔCTD datasets, approximately 1,000 particles were manually picked, followed by one round of 2D classification to generate templates for a subsequent round of auto-picking in RELION-3.0 beta. The auto-picked particles were subjected to multiple rounds of 2D classification in RELION-3.0 beta to remove "junk particles" (Zivanov *et al*, 2018). Particles in good 2D classes were extracted for initial model generation in RELION-3.0 beta (Zivanov *et al*, 2018).

The initial model was low-pass-filtered to 20 Å to serve as a starting reference for a further round of 3D auto-refinement in RELION-3.0 beta using all particles in good 3D classes. Good 3D classes were selected and iteratively refined to yield high-resolution maps in RELION-3.0 beta with no symmetry applied (Zivanov *et al*, 2018). To improve the map quality, per-particle CTF refinement and Bayesian polishing (Zivanov *et al*, 2019) were carried out in RELION-3.0 beta. The overall resolution was estimated based on the gold-standard Fourier shell correlation (FSC) cut-off at 0.143. The local resolution was calculated from the two half-maps using RELION-3.0 beta.

### Cryo-EM model building and refinement

Homology modelling was initially performed by SWISS-MODEL using the crystal structure of *Pa*NhaP as a template (PDB id: 4cz9) The model was fitted as a rigid body into the map density using MOLREP in CCP-EM suite (Burnley *et al*, 2017). After molecular replacement, manual model building was performed using COOT (Emsley *et al*, 2010). The placement of almost all TM residues was clear from the map density apart from the negatively charged Asp244 and Asp215 rotamers, which were modelled manually based on the orientation of the corresponding residues of the *Pa*NhaP structure obtained at pH 8 (PDB id: 4cz8). The final model underwent simulated annealing and NCS restraints using real-space refinement in PHENIX (Afonine *et al*, 2018). The refinement statistics are summarized in Table 1.

The sequence of the missing extracellular loop connecting TM2 and TM3 (residues 69–119) was analysed using web-based multiple secondary structure prediction methods (Jiang *et al*, 2017), which all converged to predict 2–3 beta-strand fragments encompassing three lysine residues. Since structures of NhaA have 2 beta-hairpin located in the equivalent position, loop sequences from NHE9 and NhaA extracellular loops were

therefore aligned in ClustalW and PROMALS3D (Chatzou *et al*, 2016), which revealed significant local sequence and structural homology (30–50%) around the predicted beta-strand regions. The missing loop was modelled with MODELLER (Webb & Sali, 2016) using NhaA structures as template, with the hairpins interacting laterally in antiparallel fashion covering the dimerization cleft (PDB id: 4atv). The generated loop model shows significant positional variability but only partially fills the additional cryo-EM density observed between TM2 and TM3 (Fig EV2A), suggesting that the loop is flexible whilst solubilized in detergent as seen in NhaA (Lee *et al*, 2014).

The particle stack used to reconstruct the 3.2 Å resolution 3D map from RELION was used for 3D Variability Analysis (Punjani & Fleet, 2020) in CryoSPARC v2.14.2 (Punjani *et al*, 2017). During the process, particles were subjected to a 20 Å high-pass filter to remove the micelle signal and filter the resolution to 4 Å. To analyse the linear motion from 6 "Variable Component", "simple" analysis in 3D Variability Display was used and the movies were recorded by Chimera (Pettersen *et al*, 2004).

## Multiple Sequence Alignment and contact analysis

Using NHE1-9 sequences from both *horse* and *human* as seeds, 1,332 homologous sequences from the UniProt mammalian database were retrieved using BLAST. Sequences were aligned with MAFFT (Katoh *et al*, 2019) and clustered with CD-HIT (Katoh *et al*, 2019), removing those with a similarity higher than 98%. The resulting alignment, formed by 650 representative mammalian NHE sequences, was used to compute conservation scores with ConSurf (Ashkenazy *et al*, 2016). For a wider comparison between NHE9 and cation:proton antiporters (CPA), *horse* NHE9 was used as seed to retrieve sequences from UniRef90, and the sequences from determined prokaryotic structures and their homologues retrieved by multiple BLAST were added to the alignment to re-compute ConSurf scores. The contact network of highly conserved residues (Consurf scores > 8–9) in NHE9 and inward/outward NapA structures was then analysed with in-house scripts to determine the conserved hydrophobic gates between the transport and dimerization domains.

## Cation-proton antiporter (CPA) principal component analysis (PCA)

PCA is a statistical technique to reveal dominant patterns in noisy data (Jolliffe & Cadima, 2016). Diagonalization of the covariance matrix of multidimensional data (like coordinate sets) renders the major axes of statistical variance or principal components (PCs), thus mapping complex data into a few coordinates, which contain the major trends explaining statistical variation. For proton exchangers, a set of prokaryotic exchanger structures sharing low sequence homology (~ 20%) but high structural similarity with NHE9, i.e. *Thermus thermophilus* NapA (PDB id; 5bz2, 5bz3), along with *Pyrococcus abyssii Pa*NhaP (PDB id; 4cza, 4cz8, 4cz9) and *Methanocaldococcus jannaschii Mj*NhaP1 (PDB id: 4czb), were aligned to extract the common structural fold or conserved core, mostly formed by conserved helices (576 residues per homodimer, for a total of eight homodimers). Missing residue gaps (< 5 residues) were rebuilt with MODELLER. The ensemble was aligned to the

dimerization domain of NHE9 (r.m.s.d. of 3.0 ± 1.3 Å) and used to compute the covariance matrix, i.e. the mean square deviations in atomic coordinates from their mean position (diagonal elements) and the correlations between their pairwise fluctuations (off-diagonal elements). The covariance matrix was diagonalized to obtain a set of eigenvectors or principal components (PCs), ordered according to their eigenvalues with decreasing variance, from those representing largest-scale motions up to small atomic coordinate fluctuations. Within this framework, any structure $i$ is characterized by its scalar product projections onto the conformational space defined by the major components, $PC_k$ ($k = 1,2, \ldots N$):

$$PC_k = [T_{i-0}] \cdot \cos(PC_k{}^\wedge T_{i-0})$$

where $T_{i-0}$ is the vector between the coordinates of $i$ structure and the chosen reference $0$ (NHE9, in this case), $PC_k$ is one of the major PC axis. This can classify and cluster structures, as well as extract motion information and transition pathways from them (Orellana *et al*, 2016). For the cation exchanger ensemble, the first component (PC1) captures alone ~ 55% of the structural variation associated with the inward–outward change, thus sorting out crystallographic structures along the transition pathway.

## Elastic network modelling (ENM) and transition pathway generation

Elastic network modelling (ENM) represent proteins as a simple network of residues (C-alphas) connected by elastic springs, so that diagonalization of the connectivity matrix renders 3N-6 eigenvectors or normal modes (NMs) that describe the intrinsic large-scale motions. To obtain an approximation to the intrinsic dynamics of NHE9, NMs were computed using the MD-refined potential ED-ENM, which predicts experimentally observed conformational states and their pathways (Orellana *et al*, 2010). NMs were also computed for the homologous NapA, which is the only cation exchanger for which structures of both the outward and inward states have been determined. To assess the ability of each NM to capture the inward–outward transition, its overlap with the elevator motion axis defined by PCA from the experimental ensemble (expPC1) as well as with the NapA transition was computed as in Orellana *et al* (2016). A number of mid-frequency modes (NM5 to 20) computed from NHE9 inward conformation display elevator-like movements between the dimer and transport domains that push NHE9 along the expPC1 conformational axis. For NapA, there are equivalent elevator-like NMs displaying significant overlap with NHE9 modes, expPC1 and also with the difference vector directly computed from inward and outward structures (Appendix Fig S11). To gain insight into the flexibility–rigidity pattern driving elevator-like motions in NHE9, the thermal residue fluctuations associated with the first 20 NMs were computed, finding that rigid minima "hinge" regions strictly coincide with the reported disease-associated missense mutations. To generate an approximate model for fully inward and outward NHE9 states, ENM-NMA was performed in an iterative fashion, monitoring displacements along PC1 and performing further refinement with MODELLER (Šali & Blundell, 1993; Webb & Sali, 2016). The elevator motion between the dimer and core domains ("elevator shift") was then estimated using in-house VMD (Visual Molecular Dynamics) (Humphrey *et al*, 1996) scripts.

## Model building for MD simulations

Two models of *horse* NHE9 were created by filling in all of the unmodelled loops so that they would be suitable for all-atom MD simulation, named model M1 (NHE9*) and M2 (NHE9 ΔCTD). In both cases, only protomer A was taken from the experimental structure as protomer B was virtually identical. Based on the UniProt sequence (F7B113) for *horse* NHE9, the unmodelled loop gaps in the structure were constructed as follows using MODELLER 9.13 (Šali & Blundell, 1993; Webb & Sali, 2016): (i) for the long-missing TM2-TM3 loop residues 69–119, a linker composed of flanking residues in the original sequence was inserted (changes underlined) as LRY|AT,NPH|LGN (M1) or LRY|ATA,PHL|GNA (M2); (ii) the short-missing loop residues 192–196 VYA|GQLKN|GDF (M1) or 190–196 AMV|YAGQLKN|GDF (M2) and (iii) residues 260–268 were modelled in full. In total, 100 models were generated and the best by DOPE score (Shen & Šali, 2006) was selected as the MD model for protomer A. Dimeric NHE9 was constructed by optimally superimposing the protomer A model on protomers A and B of the respective cryo-EM structure and merged using MDAnalysis (Gowers *et al*, 2016).

## All-atom, explicit solvent MD simulations

All-atom, explicit solvent simulation systems were set up in a 3:7 cholesterol : 1-palmitoyl-2-oleoylphosphatidylcholine (POPC) bilayer with a free NaCl concentration of 150 mM with CHARMM-GUI v1.7 (Jo *et al*, 2008, 2009; Lee *et al*, 2016) using the CHARMM27 force field with cmap for proteins (MacKerell *et al*, 1998, 2004), CHARMM36 for lipids (Klauda *et al*, 2010) and the CHARMM TIP3P water model. System sizes ranged from 130,949 to 160,475 atoms in orthorhombic simulation cells (109 Å × 109 Å × 107 Å to 127 Å × 127 Å × 112 Å). All ionizable residues were simulated in their default protonation states (as predicted by PROPKA 3.1; Olsson *et al*, 2011), except D215 and D244 as described below. Simulations were performed with Gromacs 2019.6 and 2019.4 (Abraham *et al*, 2015) on GPUs. Equilibrium MD simulations were performed after energy minimization and 6 stages of equilibration with position restraints (harmonic force constant on protein and lipids) with a total of ~3.75 ns, following the CHARMM-GUI protocol (Jo *et al*, 2008). All simulations were carried out under periodic boundary conditions at constant temperature (T = 303.15 K) and pressure (P = 1 bar). The stochastic velocity rescaling thermostat (Bussi *et al*, 2007) was used with a time constant of 1 ps, and three separate temperature-coupling groups were used for protein, lipids and solvent. A Parrinello–Rahman barostat (Parrinello & Rahman, 1981) with time constant 5 ps and compressibility $4.5 \times 10^{-5}$/bar was used for semi-isotropic pressure coupling. The Verlet neighbour list was updated as determined by `gmx mdrun` for optimum performance during run time within a Verlet buffer tolerance of 0.005 kJ/mol/ps. Coulomb interactions were calculated with the fast-smooth particle-mesh Ewald (SPME) method (Essman *et al*, 1995) with an initial real-space cut-off of 1.2 nm, which was optimized by the Gromacs GPU code at run time, and interactions beyond the cut-off were calculated in reciprocal space with a fast Fourier transform on a grid with 0.12-nm spacing and fourth-order spline interpolation. The Lennard–Jones forces were switched smoothly to zero between 1.0 and 1.2 nm, and the potential was shifted over the whole range and decreased to zero at the cut-off. Bonds to hydrogen atoms were converted to rigid holonomic constraints with the P-LINCS algorithm (Hess, 2008) or SETTLE (Miyamoto & Kollman, 1992) (for water molecules). The classical equations of motions were integrated with the leapfrog algorithm with a time step of 2 fs.

## Trajectory analysis

Analysis was carried out with Python scripts based on MDAnalysis (Gowers *et al*, 2016) (distances, RMSD) or Parallel MDAnalysis (PMDA) (Fan *et al*, 2019) (density). Time series of bound $Na^+$ distances to carboxyl oxygen atoms in D215 and D244 were calculated for all $Na^+$ ions as the shortest distance between $Na^+$ and either Asp OD1 or OD2 atoms. Binding of $Na^+$ ions was assessed with a simple distance criterion: any $Na^+$ ion within 3 Å of any carboxyl oxygen atom of either Asp215 or Asp244 was considered bound. Molecular images were prepared in VMD 1.9.3 (Humphrey *et al*, 1996) and UCSF Chimera (Pettersen *et al*, 2004).

## Simulations

All-atom, explicit solvent simulations were performed with models M1 and M2, different protonation states of Asp215 and Asp244 (see Appendix Table S1 for details) and different initial positioning of $Na^+$ ions (either positioned in the putative binding site or only ions in the bulk). For each system configuration, at least two independent repeats were performed by varying the initial velocities. In total, 6.68 μs were simulated.

Simulations of models M1 and M2 were stable with typical Cα root mean square deviations (RMSD) of around 3.5 Å for each protomer (see representative RMSD graphs in Appendix Fig S12A and B), typical for medium resolution structures. The secondary structure as observed in the cryo-EM structures was generally well maintained during the MD simulations except for TM3 in which the secondary structure was slightly broken in the input structures at the double-proline motif (Pro140, Pro141) and later at the extracellular end of TM3 at Pro130 (see representative secondary structure graphs in Appendix Fig S12C and D); it is possible TM3 would be further constrained with an intact TM2-TM3 loop, but in the current cryo-EM model, it showed dynamics as seen in Movie EV1.

## Coarse-grained, explicit solvent MD simulations

The model of dimeric NHE9 used for the atomistic MD simulations was coarse-grained with the MARTINI force field using ElNeDyn for secondary structure restraints (Periole *et al*, 2009; de Jong *et al*, 2013) and inserted into a 7:3 POPC:cholesterol membrane using the Charmm-GUI server (Qi *et al*, 2015). MD simulations were performed with Gromacs 2018 (Abraham *et al*, 2015) at 303.15 K and 1 bar using stochastic velocity-rescale temperature coupling (Bussi *et al*, 2007) and semi-isotropic Parrinello–Rahman pressure coupling (Parrinello & Rahman, 1981) with a time step of 20 fs. The Coulomb interactions were calculated using a reaction field with a 1.1 nm cut-off, and the van der Waals interactions were calculated using a single cut-off of 1.215 nm. Six repeats of the system were run, the longest being approximately 100 μs. The trajectories were concatenated to give a total run time of 400 μs. The membrane in the concatenated trajectory was analysed using the density analysis

module in MDAnalysis (Gowers *et al*, 2016). Densities of POPC, cholesterol and both lipids combined were generated from the 400 μs trajectory and visualized with VMD (Humphrey *et al*, 1996) as contour surfaces together with the Bendix plugin for curved helices (Dahl *et al*, 2012).

## Data availability

The coordinates and the maps for *horse* NHE9 ΔCTD and NHE9* have been deposited in the Protein Data Bank (PDB) and Electron Microscopy Data Bank (EMD) with entries PDB: 6Z3Z (http://www.rcsb.org/pdb/explore/explore.do?structureId = 6Z3Z), EMD: EMD-11067 (http://www.ebi.ac.uk/pdbe/entry/EMD-11067) and PDB: 6Z3Y (http://www.rcsb.org/pdb/explore/explore.do?structureId = 6Z3Y), EMD: EMD-11066 (http://www.ebi.ac.uk/pdbe/entry/EMD-11066), respectively[¶].

**Expanded View** for this article is available online.

## Acknowledgements
We are grateful to Magnus Claesson for critical reading of the manuscript, Björn Forsberg for advice with cryo-EM data processing and Marta Carroni at the Cryo-EM Swedish National Facility at SciLife Stockholm for cryo-EM data collection as well as Micheal Hall at the Umeå Core Facility for Electron Microscopy, UCEM and the European Synchrotron Radiation Facility (ESRF). This work was funded by grants from the Knut and Alice Wallenberg Foundation (KAW) and a European Research Council (ERC) Consolidator Grant EXCHANGE (Grant no. ERC-CoG-820187) to D.D. Research reported in this publication was supported by the National Institute of General Medical Sciences of the National Institutes of Health under Award Number R01GM118772 to O.B. and by a Wellcome Trust Investigator Award (no. 104633/Z/14/Z) to C.V.R.

## Author contributions
DD designed the project. IW and PM carried out cloning, expression screening and sample preparation for cryo-EM, and experiments for functional analysis. IW, PM and RM carried out cryo-EM data collection and map reconstruction. RM and DD carried out model building. M.L, DS and CR performed native MS. LO, CZ, RS and OB carried out MD simulations and PC analysis. All authors discussed the results and commented on the manuscript.

## Conflict of interest
The authors declare that they have no conflict of interest.

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
