## [Review Process File · The EMBO Journal]

Structure and Elevator Mechanism of the Mammalian Sodium/Proton Exchanger NHE9

Iven Winkelmann, Rei Matsuoka, Pascal Meier, Denis Shutin, Chenou Zhang, Laura Orellana, Ricky Sexton, Michael Landreh, Carol Robinson, Oliver Beckstein, and David Drew

DOI: [10.15252/embj.2020105908](https://doi.org/10.15252/embj.2020105908)

Corresponding author: David Drew (david.drew@dbb.su.se)

Review Timeline:

Submission Date:	12th Jun 20
Editorial Decision:	5th Aug 20
Revision Received:	17th Aug 20
Editorial Decision:	7th Sep 20
Revision Received:	9th Sep 20
Accepted:	10th Sep 20

Editor: Daniel Klimmeck

Transaction Report:

Dear Dr Drew,

Thank you for the submission of your manuscript (EMBOJ-2020-105908) to The EMBO Journal. Please accept again my sincere apologies for the unusual delay with the peer-review of your manuscript due to protracted referee input. Your manuscript has been sent to two reviewers, and we have received reports from both of them, which I enclose below.

As you will see, the referees acknowledge the interest and novelty of your results, although they also express a number of issues related to quality aspects of the structure presented as well as more detailed introduction of the context and discussion of the findings that will have to be conclusively addressed before they can be supportive of publication of your manuscript in The EMBO Journal.

I judge the comments of the referees to be generally reasonable and given their overall interest, we are thus happy to invite you to revise your manuscript experimentally to address the referees' comments.

Please let me know any time if you have additional questions or need further input on the referee comments.

Please see below for additional instructions for preparing your revised manuscript.

Thank you for the opportunity to consider your work for publication. I look forward to your revision.

Kind regards,

Daniel Klimmeck

Daniel Klimmeck, PhD
Editor
The EMBO Journal

Before submitting your revision, primary datasets (and computer code, where appropriate) produced in this study need to be deposited in an appropriate public database (see <https://www.embopress.org/page/journal/14602075/authorguide#datadeposition>).

The accession numbers and database should be listed in a formal "Data Availability" section (placed after Materials & Method) that follows the model below (see also <https://www.embopress.org/page/journal/14602075/authorguide#availabilityofpublishedmaterial>). Please note that the Data Availability Section is restricted to new primary data that are part of this study.

Data availability

Our journal also encourages inclusion of *data citations in the reference list* to directly cite datasets that were re-used and obtained from public databases. Data citations in the article text are distinct from normal bibliographical citations and should directly link to the database records from which the data can be accessed. In the main text, data citations are formatted as follows: "Data ref: Smith et al, 2001" or "Data ref: NCBI Sequence Read Archive PRJNA342805, 2017". In the Reference list, data citations must be labeled with "[DATASET]". A data reference must provide the database name, accession number/identifiers and a resolvable link to the landing page from which the data can be accessed at the end of the reference. Further instructions are available at <https://www.embopress.org/page/journal/14602075/authorguide#referencesformat>

- a point-by-point response to the referees' comments, with a detailed description of the changes made (as a word file).
- a word file of the manuscript text.
- individual production quality figure files (one file per figure)
- a complete author checklist, which you can download from our author guidelines (<http://emboj.embopress.org/authorguide>).
- Expanded View files (replacing Supplementary Information)

The revision must be submitted online within 90 days; please click on the link below to submit the revision online before 3rd Nov 2020.

Link Not Available

Referee #1:

Winkelman et al. & Drew working in collaboration with the Beckstein and Robinson groups have characterised horse NHE9 through a combination of methods - yeast expression, cryo-EM structure, functional assays in proteoliposomes using proton-sensitive fluorescence, biophysics of GFP-protein stability, CG-MD and homology models. The level of conclusions on ion binding sites and the putative mechanism are justified from literature and the experiments presented, and they discuss the relevance to understanding for example autism-spectrum disorders. This is fine, solid work, basically impeccable, and it should interest a large readership.

I have only few remarks:

- 1) the introduction. "Dysfunction of NHEs has been linked to diseases such as" - perhaps be a bit more specific on how various diseases are either caused by NHE dysfunction, or in other cases correlated with NHE upregulation/downregulation
- 2) the introduction: "Human disease mutations of NHE9 are linked to neurological disorders such as familial autism, ADHD and epilepsy, making NHE9 a prime drug target 8-10". This remark is a bit too easy - how could a drug targeting NHE9 overcome these disorders if they are caused by NHE9 mutations and dysfunction?
- 3) The disordered C-terminus could perhaps be addressed a bit more except to say that it is disordered as shown for the isolated NHE1 tail (refs in the paper). Please include it in the sequence alignment (suppl. fig. 1). Could interactions occur under certain functional states that are not imposed here? It would be too much to ask for more structures that address the C-terminal interactions, but it will be highly interesting to discuss putative mechanisms of auto regulation and how they may interfere with the transport models presented here, or even disease mutations.

Referee #2:

Proton/sodium ion exchangers (NHEs) are essential membrane transporters required to maintain the intracellular or intraorganellar pH, the sodium ion concentration and the volume. The manuscript describes the structure of a mammalian electroneutral proton/sodium ion exchanger (NHEs), namely NHE9 from horses, as determined by cryoEM. This is a major achievement. Many groups tried to

achieve this goal but failed because of lack of protein production or insufficient stability of the produced protein. Key for the success of the authors was a careful comparative screening of 12 NHE9s from various vertebrates, identifying the most appropriate NHE for their studies. Apart from the structure the authors show by using mass spectrometry the importance of phosphoinositides for the dimerization of NHE9 and its activity. They use molecular dynamics simulations and related methods to investigate the mechanism.

Because the methodological repertoire is very broad an appropriate evaluation of the manuscript would require experts from these various fields.

My major concerns are twofold:

-The quality of the structure, as indicated by the pdb validation report is low. In particular there are many clashes of side chains, and non-hydrogen atoms outside of the density. The authors have to improve the structure or provide an explanation for its low quality.

- Residue D244 is considered to be essential for binding the transported sodium ion. The substantial activity of variant N243A-D244A (figure 1C) is highly surprising. This fact is not mentioned nor discussed in the text. In my opinion this observation invalidates a major part of the mechanistic discussion where D244 plays the key role for the exchange activity. I suggest to use MD simulations and binding calculations to investigate whether a sodium ion still can bind near D244A.

I cannot judge the quality of the Elastic Network Modelling (ENM) and transition pathway generation.

Minor concerns:

p.3 and methods: Please explain the optimisation using *S. cerevisiae*. Were mutations induced or was the isolation procedure improved ?

p.4: hairpin extension and not hairpins extension

p.9: Why using capital letters for n-Dodecyl β -D-maltoside, Octyl glucoside, and 9-Amino-6-Chloro-2-Methoxyacridine ? These are simply chemicals.

p.14: Please clarify "was performed using carried by"

p.17: Correct spelling "Modell building"

Structure and Elevator Mechanism of the Mammalian Sodium/Proton Exchanger NHE9

Corresponding author: David Drew

We thank the referees for their considered evaluation. We have responded, as appropriate, to all queries below.

Referee #1 (Remarks to the Author):

Winkelman et al. & Drew working in collaboration with the Beckstein and Robinson groups have characterised horse NHE9 through a combination of methods - yeast expression, cryo-EM structure, functional assays in proteoliposomes using proton-sensitive fluorescence, biophysics of GFP-protein stability, CG-MD and homology models. The level of conclusions on ion binding sites and the putative mechanism are justified from literature and the experiments presented, and they discuss the relevance to understanding for example autism-spectrum disorders. This is fine, solid work, basically impeccable, and it should interest a large readership.

Thank you!

I have only few remarks:

1) the introduction. "Dysfunction of NHEs has been linked to diseases such as" - perhaps be a bit more specific on how various diseases are either caused by NHE dysfunction, or in other cases correlated with NHE upregulation/downregulation

Thank you for this suggestion. We have included more examples of NHE physiology and how they are connected to diseases.

2) the introduction: "Human disease mutations of NHE9 are linked to neurological disorders such as familial autism, ADHD and epilepsy, making NHE9 a prime drug target⁸⁻¹⁰". This remark is a bit too easy - how could a drug targeting NHE9 overcome these disorders if they are caused by NHE9 mutations and dysfunction?

We agree. This sentence is no longer included, but rather the drug targeting of NHEs is included in context of their physiological roles, which have now been explained.

3) The disordered C-terminus could perhaps be addressed a bit more except to say that it is disordered as shown for the isolated NHE1 tail (refs in the paper). Please include it in the sequence alignment (suppl. fig. 1). Could interactions occur under certain functional states that are not imposed here? It would be too much to ask for more structures that address the C-terminal interactions, but it will be highly interesting to discuss putative mechanisms of auto regulation and how they may interfere with the transport models presented here, or even disease mutations

Good point. We have now included the C-terminal tail of NHE9 with the alignment in NHE1 in Figure EV1. We have clarified in the discussion that the CTD in NHE9 lacks Ca²⁺-CaM binding sites.

We spent about an extra year to try and obtain a cryo EM structure of NHE9 with the CTD as intact as possible and/or with known interaction partners, e.g., RACK1. Given we have no hint of any map density for the CTD, we think it is unlikely the CTD in NHE9 interacts tightly with the transporter domain in the absence of binding partners, which would disagree with the proposed auto-inhibitory mechanism in NHE1. However, since NHE9 lacks Ca²⁺-CaM binding sites it is possible that the CTD regulatory mechanisms have evolved differently between NHE isoforms.

Referee #2:

Proton/sodium ion exchangers (NHEs) are essential membrane transporters required to maintain the intracellular or intraorganellar pH, the sodium ion concentration and the volume. The manuscript describes the structure of a mammalian electroneutral proton/sodium ion exchanger (NHEs), namely NHE9 from horses, as determined by cryoEM. This is a major achievement. Many groups tried to achieve this goal but failed because of lack of protein production or insufficient stability of the produced protein. Key for the success of the authors was a careful comparative screening of 12 NHE9s from various vertebrates, identifying the most appropriate NHE for their studies. Apart from the structure the authors show by using mass spectrometry the importance of phosphoinositides for the dimerization of NHE9 and its activity. They use molecular dynamics simulations and related methods to investigate the mechanism. Because the methodological repertoire is very broad an appropriate evaluation of the manuscript would require experts from these various fields.

My major concerns are twofold:

-The quality of the structure, as indicated by the pdb validation report is low. In particular there are many clashes of side chains, and non-hydrogen atoms outside of the density. The authors have to improve the structure or provide an explanation for its low quality.

Thank you for raising this point. NHE9 is very dynamic as can be seen by the large number of proline and glycine residues throughout the structure (Figure. EV5) and 3D variance analysis (Movie EV1). Indeed, we cannot apply C2 symmetry to improve map density, which we attribute to the mobility of the transport domains. Nevertheless, as can be seen from the Ramachandran plot, the fitting side-chains into cryo EM density and the expected RMSD deviation of NHE9 when embedded into a membrane by MD simulations, the NHE9 structure is of excellent quality for a protein determined at 3.2 to 3.5Å resolution (Table 1, Figure. EV2, Appendix Fig. S11).

To explain, the clashes identified in the PDB validation report were predominantly located in the flexible loop regions. We have trimmed back four residues in two different loops (49-53) and (423-427); these loop regions have no bearing on the computational analysis carried out. As can be seen in the PDB validation report, the clash score decreased in NHE9* (from 48 to 6) and NHE9-CTD (from 31 to 7). The

clash score is now close to average for a cryo EM structure modelled at this resolution as can be visualized in the PDB validation reports. The Ramachandran has also improved (Favored: from 84 to 94%, Allowed: from 15 to 6%)(Table 1).

- Residue D244 is considered to be essential for binding the transported sodium ion. The substantial activity of variant N243A-D244A (figure 1C) is highly surprising. This fact is not mentioned nor discussed in the text. In my opinion this observation invalidates a major part of the mechanistic discussion where D244 plays the key role for the exchange activity. I suggest to use MD simulations and binding calculations to investigate whether a sodium ion still can bind near D244A. I cannot judge the quality of the Elastic Network Modelling (ENM) and transition pathway generation.

Thank you for raising this point and, you are right, in hindsight some explanation in the text should have been included in our initial submission.

We have now made clearer in the text that the N243A-D244A mutation is part of the well-known critical ND motif, which is essential for ion-binding and transport. Our interpretation was that the NHE9 mutant was non-functional, and yet we did not explain why this ND mutant did not completely abolish the pH induced response by Na⁺ addition as expected. Nevertheless, since Na⁺/H⁺ antiporters have a well-documented ion-binding site (as can also be seen here by bioinformatic analysis and by structural comparison to crystal structures, especially Tl⁺-bound *PaNhaP*), the ion-binding site location and the role of the strictly conserved ion-binding aspartate for Na⁺ coordination were not in doubt *i.e.*, as such, the fact that we could still see a measurable response had no bearing on the final mechanism, see figure below.

Figure showing the MD simulations of Na⁺ binding to NHE9 (left and center panel) and the corresponding location of the Tl⁺ bound site in the bacterial homologue *PaNhaP* (right panel)

We should point out that in MD simulations when D244 was protonated we saw 0% Na⁺ binding (Appendix Table 2). We have made this control MD simulations clearer to the reader by including a new figure Appendix Figure S5; also pasted below.

MD simulation of NHE9 with deprotonated and protonated binding site residue D244. (A) Na^+ density from MD simulation (m1-10-f-2), measured in mol/l. The bulk density is ~ 150 mM (cyan). D244 in protomer B (left) was simulated in its charged (deprotonated) form and Na^+ entered the site spontaneously. D244 in protomer A (right) was simulated with a proton bound (i.e., neutral), and no density above the cutoff of 0.001 mol/l (dark blue) is detected. The membrane is omitted for clarity. (B) Top-view of the binding site with protonated D244 of the same; no density is detected near D244 and not even near the charged D215.

Either empty liposomes or a known dead-mutant, should have been appropriate controls in a proteoliposome transport assays. However, since our initial submission we have concluded that these “negative” controls inaccurately assessed the background signal in the transport assay. This became apparent to us when we could still measure the same Na^+ -induced response for a completely unrelated protein, the rat fructose transporter GLUT5. For sake of clarity, we now only show the response in the assay to rat GLUT5, rather than either empty liposomes or the NapA dead mutant (Figure 1C). We further have included a Na^+ -dependent titration of rat GLUT5 and N243A-D244A mutant, to show that the signal-to-noise is yet high enough to measure NHE9 kinetics; calculated from 3 independent protein reconstitutions (Figure 1D); pasted below.

Figure showing the Na^+ induced response of NHE9 in the proteoliposome assay as compared to a dead ND-motif mutant and the unrelated fructose transporter rat GLUT5.

Purified rat GLUT5, like NHE9, has poor stability in detergent as compared to bacterial NapA. Our current theory is that during detergent removal and reconstitution into liposomes a fraction of the mammalian proteins aggregate, which effects the integrity of some ATP-containing liposomes. We think that under a pH gradient these

liposomes burst upon NaCl addition due to osmotic pressure, which causes a proton leak and an increase in the background signal compared to empty liposomes or the dead NapA mutant. Though not ideal, we have spent considerable effort on optimizing the NHE9 proteoliposome assay, and it is unlikely we can resolve this issue in a short time frame. To put this work into broader context, however, NHEs lack working protocols using purified components and, in our hands, the functional assay development has been of equal effort as the cryo EM work. Notably, in this case, NHE9 activities are challenging to record *in vivo* due to their endosomal location.

Minor concerns:

p.3 and methods: Please explain the optimisation using *S. cerevisiae*. Were mutations induced or was the isolation procedure improved ?

Yes, we made a small C-terminal truncation based on a region of predicted disorder as described in the beginning of the methods section.

p.4: hairpin extension and not hairpins extension

Thank you, this has now been updated.

p.9: Why using capital letters for n-Dodecyl β -D-maltoside, Octyl glucoside, and 9-Amino-6-Chloro-2-Methoxyacridine ? These are simply chemicals.

Thank you, these have now been updated.

p.14: Please clarify "was performed using carried by"

Thank you, we have now fixed this typo.

p.17: Correct spelling "Modell building

Thank you, we have now fixed this typo.

Dear David,

Thank you for submitting your amended manuscript (EMBOJ-2020-105908R) to The EMBO Journal. Your revised study was sent back to referee #2 for re-evaluation, and we have received his-her comments, which I enclose below. The reviewer finds that the concerns raised have been sufficiently addressed and is now broadly in favour of publication.

Thus, we are pleased to inform you that your manuscript has been accepted in principle for publication in The EMBO Journal, pending some minor remaining issues related to formatting and data representation as detailed below which need to be addressed at re-submission.

Further, I will share additional comments from our production team during the next days to be considered.

Please contact me at any time if you have further questions.

As you might have noticed, every paper at the EMBO Journal now includes a 'Synopsis', displayed on the html and freely accessible to all readers. The synopsis includes a 'model' figure as well as 2-5 one-short-sentence bullet points that summarize the article. I would appreciate if you could provide this figure and the bullet points.

Thank you for giving us the chance to consider your manuscript for The EMBO Journal. I look forward to your final revision.

Kind regards,

Daniel

Daniel Klimmeck PhD
Editor
The EMBO Journal

- >> Add up to five keywords to the manuscript.
- >> Please introduce separate sections for 'Statistical analysis' and 'Conflict of interest'.
- >> Re-check callouts for Figure 2C in the manuscript. Fig EV3 is called out before Fig EV2C.
- >> Please format Figure EV1 such that it is limited to one page only.
- >> Dataset EV legends. All movies need their legends removed from the manuscript and zipped

with each respective movie file.

>> Manuscript order: please move acknowledgements, author contributions and conflict of interest up, after material and methods section.

>> Check bioRxiv preprint references listed for updated journal publication status.

Further information is available in our Guide For Authors:

The revision must be submitted online within 90 days; please click on the link below to submit the revision online before 6th Dec 2020.

Link Not Available

Referee #2:

The authors have addressed my previous criticism satisfactorily: the quality of the structure has been improved considerably.

A satisfactory explanation for the "activity" of the double mutant has been provided, new experiments have been performed demonstrating that the observed "activity" of the double mutant was an experimental artifact. The minor corrections have been done.

The authors performed the requested editorial changes.

Dear Dr Drew,

Thank you for submitting the revised version of your manuscript. I have now evaluated your amended manuscript and concluded that the remaining minor concerns have been sufficiently addressed.

Thus, I am pleased to inform you that your manuscript has been accepted for publication in the EMBO Journal.

Please note that it is EMBO Journal policy for the transcript of the editorial process (containing referee reports and your response letter) to be published as an online supplement to each paper. I would thus like to ask for your consent on keeping the additional referee figures included in this file.

Also in case you might NOT want the transparent process file published at all, you will also need to inform us via email immediately. More information is available here:

http://emboj.embopress.org/about#Transparent_Process

Please note that in order to be able to start the production process, our publisher will need and contact you regarding the following forms:

- PAGE CHARGE AUTHORISATION (For Articles and Resources)

[http://onlinelibrary.wiley.com/journal/10.1002/\(ISSN\)1460-2075/homepage/tej_apc.pdf](http://onlinelibrary.wiley.com/journal/10.1002/(ISSN)1460-2075/homepage/tej_apc.pdf)

- LICENCE TO PUBLISH (for non-Open Access)

Your article cannot be published until the publisher has received the appropriate signed license agreement. Once your article has been received by Wiley for production you will receive an email from Wiley's Author Services system, which will ask you to log in and will present them with the appropriate license for completion.

- LICENCE TO PUBLISH for OPEN ACCESS papers

Authors of accepted peer-reviewed original research articles may choose to pay a fee in order for their published article to be made freely accessible to all online immediately upon publication. The EMBO Open fee is fixed at \$5,200 (+ VAT where applicable).

We offer two licenses for Open Access papers, CC-BY and CC-BY-NC-ND.

For more information on these licenses, please visit: <http://creativecommons.org/licenses/by/3.0/> and http://creativecommons.org/licenses/by-nc-nd/3.0/deed.en_US

- PAYMENT FOR OPEN ACCESS papers

You also need to complete our payment system for Open Access articles. Please follow this link and select EMBO Journal from the drop down list and then complete the payment process:

https://authorservices.wiley.com/bauthor/onlineopen_order.asp

Should you be planning a Press Release on your article, please get in contact with embojournal@wiley.com as early as possible, in order to coordinate publication and release dates.

On a different note, I would like to alert you that EMBO Press is currently developing a new format for a video-synopsis of work published with us, which essentially is a short, author-generated film explaining the core findings in hand drawings, and, as we believe, can be very useful to increase visibility of the work.

Please see the following link for a representative example:

http://embopress.org/video_EMBOJ-2014-90147

If you have any questions, please do not hesitate to call or email the Editorial Office.

Kind regards,

Daniel Klimmeck

Daniel Klimmeck, PhD
Editor
The EMBO Journal
EMBO
Postfach 1022-40
Meyerohofstrasse 1
D-69117 Heidelberg
contact@embojournal.org
Submit at: <http://emboj.msubmit.net>

Corresponding Author Name: David Drew

Manuscript Number: 2020-105908